# EAGER: Asking and Answering Questions for Automatic Reward Shaping in Language-guided RL

**Thomas Carta**
Inria - Flowers team
Université de Bordeaux
`thomas.carta@inria.fr`

**Sylvain Lamprier**
ISIR Sorbonne Université
Univ Angers, LERIA,
SFR MATHSTIC, F-49000 Angers, France
`sylvain.lamprier@univ-angers.fr`

**Pierre-Yves Oudeyer**
Inria - Flowers team
Université de Bordeaux
Microsoft Research Montreal
`pierre-yves.oudeyer@inria.fr`

ISIR Sorbonne Université, Paris, France
`olivier.sigaud@isir.upmc.fr`

## Abstract

Reinforcement learning (RL) in long horizon and sparse reward tasks is notoriously difficult and requires a lot of training steps. A standard solution to speed up the process is to leverage additional reward signals, shaping it to better guide the learning process. In the context of language-conditioned RL, the abstraction and generalisation properties of the language input provide opportunities for more efficient ways of shaping the reward. In this paper, we leverage this idea and propose an automated reward shaping method where the agent extracts auxiliary objectives from the general language goal. These auxiliary objectives use a question generation (QG) and question answering (QA) system: they consist of questions leading the agent to try to reconstruct partial information about the global goal using its own trajectory. When it succeeds, it receives an intrinsic reward proportional to its confidence in its answer. This incentivizes the agent to generate trajectories which unambiguously explain various aspects of the general language goal. Our experimental study shows that this approach, which does not require engineer intervention to design the auxiliary objectives, improves sample efficiency by effectively directing exploration.

## 1 Introduction

One of the main challenges of Reinforcement Learning (RL) research is to train agents able of abstraction, generalisation and communication. Languages, be they natural or formal, afford these desirable properties [11]. Based on this insight, many papers have tried to leverage the abilities of language in RL to enable communication and improve generalisation and sample efficiency [3, 22, 12, 36]. The domain can be subdivided into language-conditioned RL (LC-RL), in which language conditions the formulation of the problem[2, 12], and language-assisted RL, where language helps the agent to learn [15, 10, 1, 9]. In the present paper, we focus on the LC-RL framework where the agent initially receives a language instruction and must act to optimise the corresponding reward function. Unfortunately, the corresponding RL algorithms are sample inefficient, especially due to the fact that the reward function is sparse when it is restricted to completing the goal.

To tackle the reward sparsity issue, one idea is to densify the reward by decomposing the general goal into sub-goals and rewarding them individually. This idea is based on a decomposition principle which

36th Conference on Neural Information Processing Systems (NeurIPS 2022).

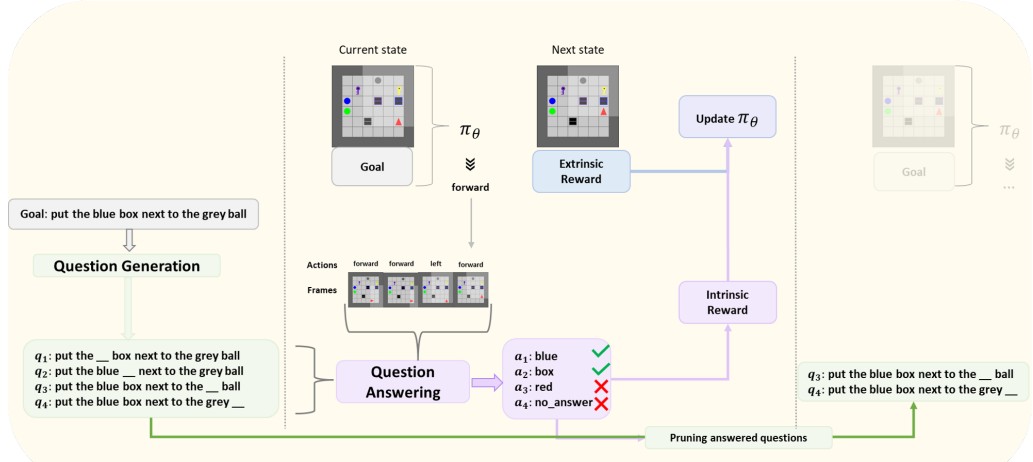

Figure 1: During training, the agent uses the goal to generate relevant questions using its question-generation module **QG**. Then, it attempts at answering them from current trajectories at each step with its question-answering module **QA**, by looking at the trajectory. When it succeeds, it obtains an intrinsic reward proportional to its confidence in its answer. Then it removes the answered questions from the list of questions. This incentivizes the agent to produce trajectories that enable to reconstruct unambiguously partial information about the general language goal, enabling to shape rewards and guide learning.

hypothesises that a general goal can be decomposed into a set of easier ones. Previous works assume a strict decomposition of the general goal into sub-goals [3, 17]. However, strict decomposition requires that that each high-level goal can be reached through an exact series of low-level policies, an assumption which fails when all the primitive actions are not known in advance. More recently, the ELLA method was proposed as a language reward shaping technique that is sample efficient and does not require a strict decomposition: it rewards interesting auxiliary objectives, without requiring rigid ordering and correspondence with sub-goals [23].

However, all these methods suffer from the need for expert input. Indeed, they require task-specific engineering for determining at least one of the following elements: the set of sub-goals or auxiliary objectives, the relevant ones and when they are achieved, and the appropriate reward. While all these points can be addressed more or less easily in simple game-like environments, they get demanding when the environment is more complex. Therefore, it seems desirable to find an alternative reward shaping approach that minimises expert involvement. We would like the agent to generate its own auxiliary objectives and create intrinsic rewards, just like a human would do.

The Natural Language Processing (**NLP**) field often suffers from the same need of expert input, e.g. for the evaluation of automatic summary tasks. In that context, some successful approaches have developed reference-less metrics [31], [29], based on question generation (**QG**) and question answering (**QA**). These techniques assess the quality of a generated text by measuring the quantity of information from the source conserved in the generated text (broadly, how well one can answer questions about the original text using the generated text). These metrics are said reference-less because they do not require a comparison with a man-made example to evaluate the quality of a text.

**Contributions** In our work, we build on these reference-less metrics to circumvent the need of expert input for generating auxiliary objectives. We adapt it and propose a novel QG/QA framework for RL called EAGER.[1] In EAGER, an agent reuses the initial language goal sentence to generate a set of questions (QG): each of these self-generated questions defines an auxiliary objective. Here, generating a question consists in masking a word of the initial language goal. Then the agent tries to answer these questions (guess the missing word) only by observing its trajectory so far. When it manages to answer a question correctly (QA) it obtains an intrinsic reward proportional to its confidence in the answer. The QA module is trained using a set of successful example trajectories. If the agent follows a path too different from correct ones at some point in its trajectory, the QA module

---

[1]Exploit question-Answering Grounding for effective Exploration in language-conditioned Reinforcement learning, see `https://github.com/flowersteam/EAGER` for access to the code.

will not answer the question correctly, resulting in zero intrinsic reward. The sum of all the intrinsic rewards measures the quality of a trajectory in relation to the given goal. In other words, maximizing this intrinsic reward incentivizes the agent to produce behaviour that unambiguously explains various aspects of the given goal.

To the best of our knowledge, EAGER is the only framework that can automatically 1) generate relevant auxiliary objectives, 2) determine their completion and 3) return the appropriate intrinsic reward. This approach only assumes the agent has access to a dataset of demonstrated behaviours associated to global language commands, which enables it to pre-train its question answering module.

Thus our work brings the following contributions:

• We create a QG/QA metric providing to an agent an information-rich measure of the quality of its trajectory given a goal.

• We propose the EAGER framework that lets the agent guide its own learning process by generating auxiliary objectives and producing intrinsic rewards without requiring any expert intervention.

• We show that EAGER retains the good properties of ELLA without requiring task-specific expert knowledge, by leveraging properties of language.

• We experiment EAGER with the BabyAI platform [7]: we compare our approach against ELLA (SOTA on BabyAI) and RIDE (a non-language based reward shaping approach using intrinsic motivation), showing its robustness and sample efficiency. Furthermore, although we use example trajectories to train the QA, their use is much more parsimonious than training an agent using behavioural cloning [7], as we show in Appendix C.

## 2  Related work

**Language-conditioned RL.**    We place our work in the LC-RL setting, where an agent learns a policy to execute language commands [24, 7, 20, 21]. We reuse the BabyAI platform [7], widely used in this domain as it enables to decouple exploration challenges from perception challenges. It uses a synthetic language exhibiting interesting combinatorial properties with possible conjunction of properties, and procedural generation to avoid overfitting [8]. Here, we consider *instruction following* agents which receive external instructions and rewards [14, 5, 17].

**Language as an abstraction in hierarchical RL.**    Several approaches leverage language for abstraction in hierarchical RL. One approach uses language for training a low-level instruction-following policy, then learns a high-level policy that generates the sequence of low-level goals [17]. Another one explicitly decomposes high-level tasks into low-level ones as in policy sketches [3]. ELLA [23] also uses language to decompose high-level tasks but relaxes the strict decomposition constraint by replacing sub-goals with auxiliary objectives. We adopt the same flexible framework in our approach but automatise the decomposition.

**Language for exploration and reward shaping.**    Reward shaping is a form of guidance that supplies additional rewards to the agent to direct its learning process. Among approaches studying how language can shape rewards and exploration, LEARN [12] proposes to map intermediate natural language instruction to intermediate rewards. Similarly, [35] enables reward shaping using natural language through a narration-guided method. The high-level tasks are decomposed into low-level tasks and rewarded using narration. ELLA [23] is positioned in the same paradigm but with fewer assumptions about the environment or the structure of the task.

Other approaches assume that an oracle provides language descriptions of environment states which are used as state abstraction to generate novelty intrinsic rewards and guide exploration [25, 34]. This extends classical approaches to intrinsically motivated exploration [4], such as count-based methods [6] or RIDE [28]. Here, we also use language to generate intrinsic rewards, but we do not need language descriptions of states. Besides, other forms of intrinsic motivation systems, such as IMAGINE [10], first learn language-parameterized reward functions through interaction with a social peer, then autonomously use language to generate diverse and novel goals, using the learned reward function for self-supervision.

**Asking questions in RL.** Beyond reward shaping, some methods consider agents that use language to ask questions to external knowledge sources. In QWA [36], questions are used to identify sub-tasks and prune irrelevant actions. In AFK [19], questions are used to obtain world knowledge that helps completing tasks.

**Natural Language Processing** One of the sources of QG/QA methods is the thriving field of question generation from natural language processing and information retrieval [16]. Our approach is inspired from text generation methods where QG and QA are used to measure the quality of a generated text without using a human reference [31, 29].

## 3 Problem statement

We set ourselves in the standard framework of LC-RL with an augmented Partial Observation Markov Decision Process [33] $\mathcal{M}$ defined by the tuple $(\mathcal{S}, \mathcal{A}, \mathcal{Z}, \mathcal{T}, \mathcal{O}, \mathcal{G}, \mathcal{V}, \mathcal{I}, \mathcal{R}, \gamma)$, with $\mathcal{S}$ the state space, $\mathcal{Z}$ the observation space, $\mathcal{A}$ the actions space, $\mathcal{G}$ the goal space, $\mathcal{V}$ the vocabulary of goal instructions. $\mathcal{O}$ stands as the observation function $\mathcal{O} : \mathcal{S} \rightarrow \mathcal{Z}$, which maps states to the observations space. $\mathcal{I}$ is the instruction function $\mathcal{I} : \mathcal{G} \rightarrow \mathcal{V}^{isize}$, which maps any goal in $\mathcal{G}$ to the set of language instructions, which correspond to sequences of $isize$ symbols (the empty symbol $\epsilon$ belongs to $\mathcal{V}$ to allow variable instruction sizes). $\mathcal{R}$ is a goal-conditioned state action reward function, with $\mathcal{R} : \mathcal{S} \times \mathcal{A} \times \mathcal{G} \rightarrow [0, 1]$ the extrinsic reward received for some goal $g \in \mathcal{G}$ and $\gamma$ the discount factor. For simplicity, we note in the following $r_t^g = \mathcal{R}(s_t, a_t, g)$ as the reward obtained at step $t$ of any episode with goal $g$.

At each time step $t$, the agent receives an observation $o_t \in \mathcal{Z}$ following the observation function $\mathcal{O} : \mathcal{S} \rightarrow \mathcal{Z}$ and selects an action $a_t \in \mathcal{A}$ to reach a goal $g \in \mathcal{G}$, expressed by $\omega^g = \mathcal{I}(g)$. $\mathcal{T} : \mathcal{S} \times \mathcal{A} \rightarrow \mathcal{P}(\mathcal{S})$ is the transition function. Using RL, we search for an optimal goal-conditioned policy $\pi^*$, such that $\pi^* : \mathcal{S} \times \mathcal{V}^{isize} \rightarrow \mathcal{A}$ maximises the discounted expected return $R_t^{\pi_g} = \mathbb{E}_\pi[\sum_{k=0}^{T} \gamma^k r_{t+k+1}^g]$. We consider in this work sparse reward problems, where $r(s, a, g)$ returns 1 for any state $s$ such that $d(s, g) \leq \epsilon$, for a given distance function $d$ and a specified threshold $\epsilon$, and 0 otherwise. Moreover, we assume a limited number of steps at most $H$ steps. These two conditions result in a hard exploration problem.

To deal with those problems, many methods aim at densifying rewards, by focusing on auxiliary objectives during training, whose accomplishment can help the agent to reach the goal $g$ at hand. In previous work, such as ELLA [23], the selection of relevant objectives required the intervention of an expert (in the form of expert annotations and example trajectories), which can be problematic because new expert trade-offs have to be established for each new environment. Thus, an automated way must be found to recover the relevant auxiliary objectives, measure their completion, and associate the appropriate reward.

Rather than relying on expert knowledge for defining auxiliary objectives, we assume that we have access to a set of trajectories of successful examples coupled to their respective instructions $\{(\tau_0, \omega_0^g), ..., (\tau_n, \omega_n^g)\}$, where $\tau_n = (o_i, a_i)_{i \in [|0, k|]}$ with $k$ the number of steps. For any goal instruction $\omega^g$, we consider a function $f$ that aims at generating a set of auxiliary goals of $g$, such that all $g' \in f(\omega^g)$ belong to $\mathcal{G}^g$, with $\mathcal{G}^g \subset \mathcal{G}$ the set of goals that help training the agent towards $g$. Then, for a trajectory $\tau$ and any $g' \in f(\omega^g)$, a function $h$ determines the probability $h(\tau, g') \in [0, 1]$ that the auxiliary objective $g'$ has been achieved. We train $h$ using the example demonstrations such that we can leverage its generalisation abilities to use $h$ for unseen trajectories.

Please note that Behavioural Cloning methods also rely on the exploitation of expert trajectories, by learning the policy $\pi_{BC}$ that maximises the log-likelihood $\mathcal{L}_{BC}$:

$$\mathcal{L}_{BC} = \sum_{i=0}^{n} \log P(\tau_i) \text{ where } \log P(\tau) = \sum_{j=0}^{k} \log(\pi_{BC}(a_k|o_k)\mathcal{T}(s_{k+1}|s_k, a_k)).$$

However, the latter uses demonstrations to train an agent to copy an expert based on a data set of example trajectories. This technique is only effective when the agent is close enough to the demonstrated examples. A large number of demonstrations are therefore required to cover the space of states that the agent may encounter. Methods such as GAIL [18] can partially circumvent this problem by forcing the agent to stay on known trajectories, but this impairs generalisation. Our method is much more parsimonious in the use of such examples: because it generalises well, the

agent can receive intrinsic rewards even in areas not encountered in the expert demonstrations (see Appendix C).

# 4 Method

In this section, we first introduce EAGER, our automated reward shaping method based on QG/QA, then we present a practical implementation in the context of the BabyAI benchmark. Figure 1 provides a graphical overview of EAGER.

## 4.1 EAGER

We need an automatic evaluation method that is fine-grained enough to rank various trajectories depending on a language instruction. But various successful policies can generate a set of valid trajectories for the same goal. It could be deceiving to rank them based only on their final results, as an overly complex trajectory seems intuitively worse than a simpler one even if the result is the same.

To address this issue, the EAGER framework consists of an agent learning module, a question generation module $QG$ (automatic, but not learned) and a learned question answering module $QA$. .These two modules fulfil the role of the functions $f$ and $h$ defined in Section 3. EAGER is inspired from works like QuestEval [31] and Data-QuestEval [29], developed for natural language generation. For instance, for abstractive summarization, by generating questions from the original text (QG) and trying to answer them using the summary QA, this method measures the quantity of information shared between both texts.

In our work, we draw the analogy with the NLP task with the goal replacing the original text and the trajectory replacing the summary. We use the QG/QA system to verify that a trajectory contains the same level of information as the language instruction, meaning that the goal is contained in the trajectory. As the goal can be contained in a lot of different trajectories, we also favour simple trajectories. If one can easily answer the question, that means the trajectory is simple.

**The QG module** We assume that the goal linguistic instruction $\omega^g$ of $g \in \mathcal{G}$ is a highly expressive language instruction (e.g *Put the red ball next to the blue box*, *open the red door*, ...), containing by themselves enough world knowledge to generate questions by masking words. Thus, the QG module returns a list of $k$ questions $QG_k(\omega^g)$ that can be seen as auxiliary objectives. Each question is formed by masking one word in the linguistic instruction. Crucially, the choice of words to mask can be done automatically without any expert knowledge of the task, or the environment, for instance masking all nouns and adjectives. These questions can be seen as auxiliary objectives guiding the agent during training. Besides, being formulated in natural language, these auxiliary objectives are easily interpreted.

**The QA module** Let $\tilde{A}$ be the set of possible answers generated automatically from the list of tokens masked by the QG. Thus, $\tilde{A}$ contains $(q, a^*)$ pairs of questions and expected answers. The QA module returns the probability for all $\tilde{a} \in \tilde{A}$, that $\tilde{a}$ answers question $q$, given a trajectory $\tau_t$, where $\tau_t = (o_i, a_i)_{i \in [|0, t|]}$ are state-action pairs and $t$ the time step. We note $\tilde{a}^* = QA(q, \tau_t)$ the answer greedily generated from the module and $QA(\tilde{a}^* \mid q, \tau_t)$ the associated probability. The auxiliary objectives from the QG are considered achieved when the QA answers them successfully. In our work, the QA module is pre-trained using full example trajectories generated by a bot, see Section 5.1, without any type of annotations to guide it. Besides, at time step $t$, there is no guarantee that the trajectories contain enough information to correctly answer a question. Thus we include a *«no_answer»* token in $\tilde{A}$ to prevent the QA from answering correctly by chance. Moreover, since the QA module takes the whole trajectory, once it has answered a question, it can also answer it at the next step. To avoid giving a reward that does not have direct link to the current step, every time a question $q$ is answered, it is removed from the set of questions: We note $\mathcal{Q}_t$ the active set of questions, the initial set is $\mathcal{Q}_0 = QG_k(\omega^g)$ and once a question is answered, we apply $\mathcal{Q}_t \leftarrow \mathcal{Q}_{t-1} \backslash \{q\}$.

**Architecture of the QA** The QA is used to compositionally chain low-level tasks. To do this, it relies on the episodic transformer [27]. This architecture uses multimodal transformers (over language, visual observation, and a list of actions) that have access to the full episode. For any time

step $t$ and any question, a $(x_{1:L}, v_{1:t}, a_{1:t})$ tuple is given as input to the QA module. The language input $x_{1:L}$ is the question, it is a sequence of $L$ tokens with $x_i \in \mathbb{N}$. The visual input $v_{1:t}$ is the list of observations $v_t \in \mathbb{R}^{W \times H \times 3}$. Finally, the action input $a_{1:t}$ is a list of discrete actions. As output, the network returns the probability distribution over the set of possible answers $\tilde{\mathcal{A}}$.

**Intrinsic reward**    The main difficulty in ranking trajectories is that many trajectories can share the same description and thus can answer questions correctly. However, a human assessing several trajectories, with the same extrinsic reward, relatively to the same goal would prioritise the simplest one. To account for this, we make the reward proportional to the confidence in the answer at time step $t$: $QA(\tilde{a}^* \mid q, \tau_t)$. An overly complex trajectory being harder to understand for the QA, the answer should be given with less confidence, so it should obtain less reward than a direct and clear trajectory, even with the same number of correct answers.

In this paper, we keep the same desired properties for the shaped reward function as the ones given in ELLA: the reward shaping should not change the optimal policy that prevails before the reward shaping (policy invariance), and the reward should encourage sample efficient exploration based on auxiliary-objectives. Using intrinsic rewards, we modify the global reward from $r$ to $r'$ through a policy invariant transformation [26], which ensure that the new policy $\pi^*$ is optimal for both $\mathcal{M}$ and $\mathcal{M}' = (\mathcal{S}, \mathcal{A}, \mathcal{T}, \mathcal{G}', \gamma)$. To do so, we ensure that only successful trajectories get the same return with or without the reward shaping by substracting the shaped reward at the final time step $N$ of a successful trajectory: $r'_N = r_N - \sum_{t \in T_{\mathbb{S}}} \gamma^{t-N} r'_t$ with $r'_t = \lambda \sum_{q \in \mathcal{Q}_t} \sum_{\tilde{a}^* \text{ is correct answer}} QA(\tilde{a}^* \mid q, \tau_t)$, where $T_{\mathbb{S}}$ is the set of time steps where a bonus is applied. $r_t$ is the reward given at time step $t$, in the case of sparse reward studied here $r_t \neq 0$ only if $t$ corresponds to the last step of a successful trajectory. In ELLA Mirchandani et al. [23] the authors prove that this transformation is policy invariant. Thus, as long as the policy produces unsuccessful trajectories, the agent is guided by the shaping reward. Then once it has learn to successfully complete an instruction, the shaping reward is substracted at the last step and the agent improves using only the extrinsic reward. Further details are provided in the appendix.

The QA system is pre-trained with successful trajectories, which prevents reward hacking. Indeed, if the agent individually completes the auxiliary objectives without a meaningful trajectory, the QA does not consider the trajectory meaningful and answers: "no_answer", preventing the agent to get rewarded.

Using the above notations and concepts, we can define a metric to measure the adequacy of a trajectory to a goal, that corresponds to our cumulative intrinsic reward over a trajectory $\tau$ of length $N$, up to a $\lambda$ factor:

$$m_{QG/QA}(g, \tau) = \sum_{t=0}^{N} \sum_{q \in \mathcal{Q}_t} QA(\tilde{a}^* \mid q, \tau_t) \, \mathbb{I}[\tilde{a}^* \text{correct answer to q}]. \tag{1}$$

**Algorithm**    Our algorithmic procedure is given in Algorithm 1. At the beginning of an episode, the QG takes the goal and returns a set of questions related to it. Then the QA module is applied at each step over the active set of questions $\mathcal{Q}_t$. When a question is answered correctly, the shaped reward function returns a bonus $\lambda \, QA(\tilde{a}^* \mid q, \tau_t)$, where $\lambda$ is a scaling factor, to the agent and the answered question is removed from the set of active questions.

Then we tune $\lambda$ to ensure that no unsuccessful trajectories can get more reward than a successful one from the optimal policy $\pi^*$. The higher bound for the shaped reward of unsuccessful trajectories is $\lambda \, k$ and the lower bound for the reward of a successful trajectory is $\gamma^H \, r_H$, where $k$ is the number of questions generated by the $QG$ and $H$ is the maximum length of an episode. Thus we obtain

$$\lambda < \frac{\gamma^H \, r_H}{k}. \tag{2}$$

Note that by making a less conservative hypothesis, i.e. assuming in the worst case the successful trajectory takes N<H steps, we could obtain a higher $\lambda$ leading to faster learning [23].

---

**Algorithm 1** Automatic auxiliary goal generation and reward shaping using EAGER

---

**Input:** $\theta_0$ initial policy parameters, $\lambda$ bonus scaling factor, ENV the environment and OPTIMISE an RL optimiser

    **for** k=0,..., $n_{step}$ **do**
        $\omega^g, o_0, \text{done}_0 \leftarrow$ ENV.$reset()$
        $\mathcal{Q}_0 = \{q_1, ..., q_k\} \leftarrow QG_k(\omega^g)$
        $t \leftarrow 0$
        **while** $\text{done}_t$ not True **do**
            $a_t \leftarrow \pi^{\theta_0}(o_t)$
            $o_{t+1}, r_t, \text{done}_{t+1} \leftarrow$ ENV$(a_t)$
            $r'_t, \mathcal{Q}_{t+1} \leftarrow$ QA_SHAPE$(\mathcal{Q}_t, \tau_t, r_t)$
            **if** $\text{done}_{t+1}$ is True **then**
                $N \leftarrow t$
                $r'_N \leftarrow$ NEUTRALISE$(r'_{1:N})$
            **end if**
        **end while**
        Update $\theta_{k+1} \leftarrow$ OPTIMISE$(r'_{1:N})$
    **end for**

**function** QA_SHAPE$(\mathcal{Q}_t, \tau_t, r_t)$
    **for** $q$ in $\mathcal{Q}_t$ **do**
        $\tilde{a}^* \leftarrow QA(q, \tau_t)$
        **if** $\tilde{a}^*$ is correct answer to q
**then**
            $r'_t = r_t + \lambda\, QA(\tilde{a}^* \mid q, \tau_t)$
            $\mathcal{Q} = \mathcal{Q} \backslash \{q\}$
        **end if**
    **end for**
    **return** $r'_t, \mathcal{Q}$
**end function**

**function** NEUTRALISE$(r'_{1:N})$
    $r'_N \leftarrow r'_N - \sum_{t \in T_{\mathbb{S}}} \gamma^{t-N} r'_t$
    **return** $r'_N$
**end function**

---

## 4.2 A particular instance of the method in the BabyAI framework

We now explain how to adapt the EAGER method to train an RL agent in BabyAI, a language-conditioned environment where the agent has a limited number of steps to complete a language goal. In this environment the agent receives a reward if and only if it finishes the task successfully.

The BabyAI benchmark contains tasks with highly expressive language instructions e.g *Put the red box next to the green key*, ...). Thus they are rich enough to generate questions by masking words. In practice, we mask nouns and adjectives: this form of QG is very simple and can be automated using standard NLP techniques, thus it does not require expert knowledge. For instance, for the goal *Put the red ball next to the blue box*, using the token *«question»* as a mask we generate 4 questions among which *Put the «question» ball next to the blue box*.

The environments in our experiments are partially observable. Thus, our agent takes sequences of observations $(o_1, o_2, ..., o_t)$ as inputs of a recurrent network [13].

# 5 Experiments

## 5.1 Experimental settings

We use the BabyAI [7] platform to run our experiments. This platform relies on a gridworld environment (MiniGrid) to generate a set of complex instructions-following environments. It has been specifically designed for research on grounded language learning and related sample efficiency problems. The gridworld environment is populated with several entities: the agent, boxes, balls, doors, and keys of 6 different colors. These entities are placed in rooms of $8 \times 8$ tiles that are connected by doors that could be locked or closed. The agent can do 6 primitive navigation actions such as `forward`, `toggle`, `pick up` to solve the language instruction (for instance `Pick up the red box`). It only has access to partial observations of its environment inside which irrelevant objects are randomly added. Moreover, the observations are in a symbolic space using a compact encoding, with 3 input values per grid cell,[2] $8 \times 8 \times 3$ values in total. When the agent completes the task after $N$ steps, it receives the reward $r_N = 1 - 0.9\frac{N}{H}$, where $H$ is the maximum number of steps. During training, all rewards are scaled up by a factor of 20 to ensure a good propagation of the rewards. If the agent fails, the reward is 0. We focus our tests on tasks of varying complexity: PutNextTo, Unlock and Sequence. The task can also take place in one room `local` or two rooms `Medium`.

To train the QA module through supervised learning, we build a dataset of example trajectories associated to language goals using a bot provided in BabyAI, then we generate related questions and

---

[2]3 integers: one representing the shape of the object, one its color, and one its state. For instance, (4, 1, 1) represents a closed green door

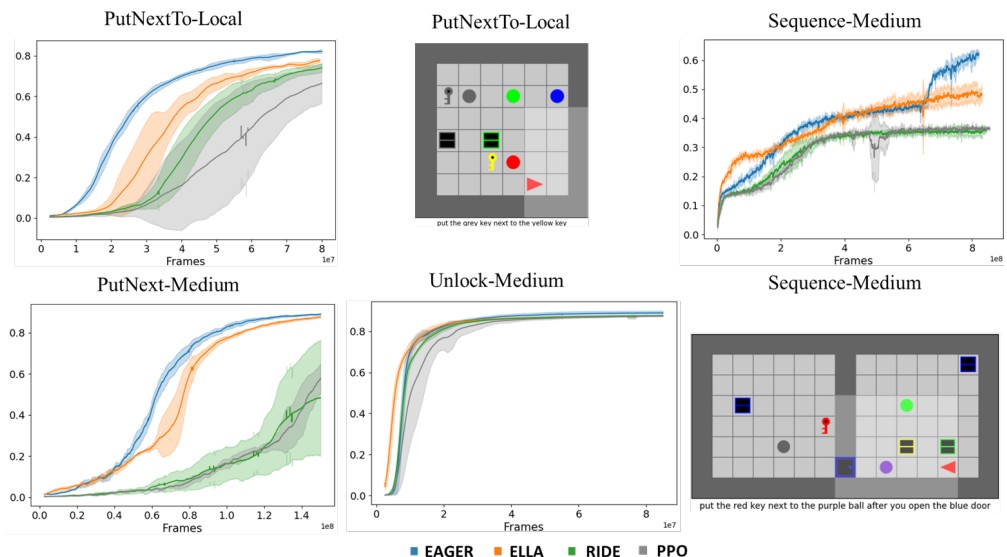

Figure 2: Average reward for EAGER and baselines for four tasks, with error regions to indicate standard deviation over four random seeds. For the PutNext-Local and Sequence-Medium tasks, we give an example of possible tasks with environment at time step $0$.

answers. To obtain a QA that can operate on various tasks, we use a mix of the PutNextTo, PickUp, Open, and Sequence tasks for generating training trajectories. This dataset is only used to train the QA, not to bootstrap policy learning. During training, we give as input of the QA the full trajectory — the list of observations and actions — and all the questions generated by the QG and we use the cross-entropy loss over its output distribution to update it. To train the QA to answer: "no_answer" and prevent it from guessing the answer by chance, we randomly associate some trajectories and questions from unrelated goals. For instance, we associate the trajectory from the goal: *take the red box next to the blue ball* and associate it with the question: *take the red «question» next to the blue key* generated from the goal *take the red box next to the blue key*. The QA must learn to pay attention to all the details of the question. It has to see that the trajectory describes an agent taking the *red box* but placing it next to an object other than the one in the question. Thus the good answer to the question is *no_answer* and not *box*.

Moreover, we empirically show in Section 5.4 that the QA is more efficient when it learns from a broad distribution of trajectories for similar tasks. The intuition behind such behaviour is that a QA trained on a narrow distribution of successful bot-generated trajectories will not recognise the noisy trajectories of the agent when it starts training. Thus the QA will too often answer "no_answer", resulting in no intrinsic reward and hurting the reward shaping efficiency. To produce a wider distribution using the given procedural bot, we force the bot to take with a certain probability a random action at each time step and only keep successful noisy trajectories as training examples. More details on the QA pre-training are given in Appendix B.1. All the tasks we use for training the QA and the agent are summarised with examples in Appendix A.

To evaluate our reward shaping framework, we use the Proximal Policy Optimization (**PPO**) algorithm [30], but our reward shaping method is algorithm agnostic. We compare our framework to PPO without reward shaping, ELLA and RIDE. RIDE [28] is an exploration method that does not use language and addresses sparse reward problem by rewarding impactful change in state. We use Nvidia Tesla V100 with 10 cores to train our model and we use 4 seeds in each experiment.

Figure 2 presents learning curves for EAGER, ELLA and RIDE across 4 environments. Table 1 describes the assumptions and the type of expert knowledge required by the three reward shaping methods. It clearly appears that EAGER requires less expert human intervention than ELLA.

Table 1: All the assumptions and expert knowledge required for RIDE, EAGER, and ELLA

| Method | Number of expert demonstration per task | Human Expert Intervention | Automated parts done by the agent |
|---|---|---|---|
| RIDE | 0 | None | Determining if a new state is impacting |
| EAGER | 7.500 noisy bot trajectories (see supplementary *Wide distribution of trajectories*) | Determining what words are nouns or adjectives | Determining relevant auxiliary objectives
Determining auxiliary objectives completion
Determining auxiliary objectives associated reward |
| ELLA | 15.000 bot trajectories | Determining the class of relevant auxiliary objectives
Determining auxiliary objectives associated reward | Determining relevant auxiliary objectives among the predetermine class
Determining auxiliary objectives completion |

## 5.2 How does EAGER perform when sparsity increases?

In the PutNextTo and Unlock tasks, EAGER obtains results better than ELLA (SOTA in BabyAI) without using expert knowledge. It also performs significantly better than RIDE for the tasks PutNext and Sequence and slightly better for Unlock. The better performance of EAGER with respect to RIDE is not surprising as the EAGER agent receives some indications based on example trajectories through the QA module.

For Unlock-Medium, EAGER overcomes a bottleneck. The general goal being *Open the «colour» door*, the agent has to first pick up the key of the corresponding *colour* before reaching the door to open it. ELLA rewards picking up keys, via the PICK low-level instruction chosen via expert knowledge. EAGER reaches better performance (see the statistical test in Appendix C) without the need for expert knowledge. Moreover, although this is not its main purpose, EAGER gets a similar or better sample efficiency for most tasks.

## 5.3 How does EAGER perform with a sequence of tasks under a temporal constraint?

The Sequence task adds a temporal constraint by chaining two tasks using 'before' or 'after' together with a high number of instructions (over 1.5M instructions in comparison with PutNext-Medium with 1440 instructions). Moreover both EAGER and ELLA decompose the goal into auxiliary objectives. This decomposition does not retain the temporal constraint, there is no notion of doing one auxiliary objective before another.

Our tests show that EAGER retains strong performance, doing better than RIDE and ELLA. The slow progress of EAGER at the beginning can be attributed to the time when the agent is not good enough to efficiently trigger an intrinsic reward signal from the QA module. Indeed, at the beginning, trajectories are noisy and it is more difficult for the QA to exploit a trajectory with more rooms. As a result, it over-responds "no_answer" leading to a lesser intrinsic reward.

## 5.4 Is EAGER robust to QA performance?

At first glance, the reliability of the QA looks crucial to our method. However, the QA could be difficult to train in some environments e.g. if you want the QA to learn to answer in a large set of answers from a small number of example trajectories. This is why we tested the robustness of our method using the PutNextTo-Local task with two metrics: the success rate SR of the QA after pre-training and the distribution of example trajectories. For the former, we take the same QA at different training epochs and we determine its SR over a test dataset, then we train the agent using the reward shaping provided by this QA. Figure 3(left) shows the robustness of our method with agents that display similar training curves as soon as $SR > 0.56$.

In Figure 3(right), we plot the SR of the QA when training the agent. Initially, the QA with a $SR \leq 0.56$ at pre-training time tends to have a higher SR. Indeed, the distribution of answers is less peaked and answers are often correct by chance. On the opposite, the QA with a $SR > 0.56$ answers "no_answer" and obtains no reward. However, in this case, the agent learns faster because it only receives a reward for meaningful answers. For the QA with a $SR \leq 0.56$, the SR along training first grows then decreases. First, the agent is biased by the intrinsic reward to follow a path that improves

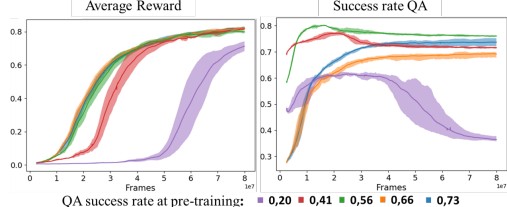
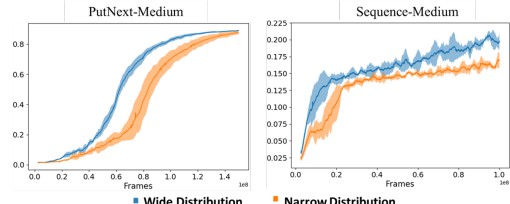

Figure 3: Average reward (left) and success rate of the QA (right) for the PutNextTo-Local tasks. The agent are trained with QA having different success rate after pre-training.

Figure 4: Average reward in tasks PutNextTo and Sequence for two distributions of bot trajectories used to train QA. A narrow and a wide distribution where noise is added to bot trajectories

the SR, but once the agent learns to complete the trajectory leading to the extrinsic reward, the SR converges to pre-training SR.

As explained in Section 5.1, we added noise to trajectories generated by the bot to compensate for a too narrow trajectory distribution. Figure 4 shows training curves for two environments for QA trained on wide trajectory distribution (WD) and narrow distribution (ND). The reward shaping method trained on (WD) learns faster because they efficiently reward the agent early in training.

### 5.5 How do design choices on the QA module affect EAGER's performance?

In Section 4.1 we made two choices for the QA module and the associated intrinsic reward: first we added a "no_answer" response, second we rewarded each answer by the confidence the agent had in its own answer. To verify the influence of these choices over EAGER's performance, we use the PutNextTo task to compare EAGER against "EAGER \no_answer", "EAGER Simple", and "EAGER Simple \no_answer". "Simple" means that the agent received a binary intrinsic reward (1 for a good answer, 0 otherwise) and "\no_answer" means we suppressed the "no_answer" solution. Figure 5 gives the results of these ablations. We can see that both the use of "no_answer" and the non-binary reward independently boost sample efficiency.

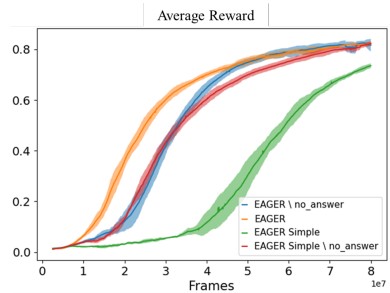

Figure 5: Average reward in PutNextTo with various ablations of the QA module.

## 6 Conclusion

In this work, we have proposed to leverage the abstraction and generalisation properties of language to build an automatic reward shaping method in the context of long horizon and sparse reward tasks. Our learning agent generates its own questions from the goal and rewards itself for correctly answering them, resulting in an efficient curriculum over auxiliary objectives. This is to be contrasted with ELLA [23] where expert knowledge is required for choosing auxiliary objectives. Besides, we do not call upon an oracle for getting linguistic description of environment states as in [25].

**Limitations and Future Work**  EAGER assumes the QA system was pre-trained using a pre-existing set of example trajectories. Next steps will consist in investigating how to remove this limitation, e.g. by implementing autotelic strategies based on QG/QA learned online. Besides, in this work we tested our method on BabyAI, a 2D environment with synthetic language. In the future, we would like to consider a more complex language, generating more complex questions than the one obtained by masking, and testing our method on more realistic environments with true human instructions, as in the ALFRED dataset [32].

## Acknowledgments and Disclosure of Funding

This work benefited from the use of the Jean Zay supercomputer associated with the Genci grant A0091011996, as well as from the ANR DeepCuriosity AI chair project.

This work has been supported by the Horizon 2020 PILLAR-robots project (grant number 101070381)

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
