This supplementary material provides additional results and discussion, as well as implementation details.

- Section A summarises the different tasks and the assumption used in RIDE, EAGER, ELLA.
- Section B gives more details about training of the QA module and the agent. It also includes explanations of how we built the training data set for the QA module.
- Section C gathers several results on EAGER: comparison with behavioural cloning, generalisation capacity of QA, robustness results of EAGER...
- Section D contains a commented version of the EAGER algorithm.
- Section E summarises hyperparameters.

## A   Tasks description and assumptions used for the different method of reward shaping

Table 1 describes the tasks used in the experiments with an example and if it has been used to train the QA module or the agent. The Unlock and Open tasks have the same type of instructions, the agent can nevertheless see the difference because in the Unlock task, the door is a solid square where in the Open task, the door is just a border.

Table 1: Tasks description, ✓ means that the task is used for training the QA resp. the Agent

| Task | Explanation | Example | Train QA | Train Agent |
|------|-------------|---------|----------|-------------|
| PutNextTo | put an object next to another | *put the purple ball next to the blue key* | ✓ | ✓ |
| PickUp | pick up an object | *pick up a red box* | ✓ | |
| Open | open a door (does not require a key) | *open the green door* | ✓ | |
| Unlock | open a door using the key of the same colour | *open the green door* | | ✓ |
| Sequence | sequence of two of the previous tasks: task 1 before/after task 2 | *put the blue key next to the red box before you open the grey door* | ✓ | ✓ |

## B   Training details

In this section we explain how we create the QA training data set, the architecture of the agent used for RL training, and give more details on the baselines.

### B.1   QA architecture and pre-training

**QA architecture**   The QA architecture is based on the Episodic Transformer architecture [6] depicted in Figure 1. Using multimodal transformers, the QA can direct its attention over certain observations in the trajectory about the words used for the question and the previously taken action. Thus we can train the QA over the full trajectory and use it on partial trajectories (up to time step $t$) at test time.

**QA training data set**   We train the QA with a mix of 4 tasks: Open-Large, PickUp-Large, PutNextTo-Local, and Sequence-Medium. We use a bot provided with the BabyAI platform that procedurally solves the environment. Using it, we generate 7500 example trajectories for each task. We use a mix of tasks to push the QA to leverage the compositionality of language. Indeed, the Sequence task is created by putting in sequence two tasks from Open, PickUp, and PutNextTo. Compared to QA trained on an individual task, our QA needs fewer examples per task. Thus the QA can use the environment with the goal *put the blue key next to the red box then open the grey door* to ground the instruction both for PutNextTo and Open.

**Adding "no_answer" questions**   To train the QA to respond: "no_answer" and prevent it from hazard-guessing the answer, we randomly associate certain paths and questions from unrelated objectives. To generate these questions, for each new trajectory generated by the bot, we take a goal among the last three used. If this goal differs from the goal used for the trajectory, we use it to

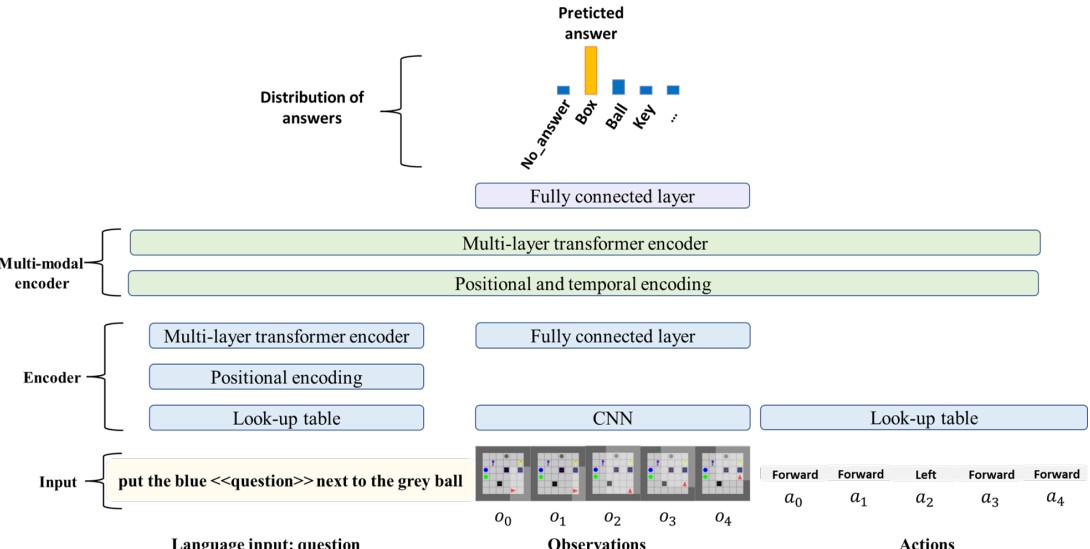

Figure 1: The QA architecture is based on the architecture of the Episodic Transformer [6]. To predict the correct answer the QA model received a natural language question, visual observations, and actions generated by the agent from the beginning of an episode (see Figure 3). Here we show an example that corresponds to the $5^{th}$ time step of an episode. After processing the question with a transformer-based language encoder, embedding the observation with a convolutional neural network CNN and passing actions through a look-up table, the agent outputs the correct answer. During training, we use full trajectories. At test time, we ask all the questions in the active set of questions using the current trajectory.

create questions that are associated with "no_answer". To avoid biasing the dataset with too much of these questions, we randomly keep some of the generated "no_answer" questions. The probability to keep a question depends on the number of words in common between the goal corresponding to the trajectories and the random goal. If these two goals share a lot of words in common, it is harder for the QA to see that it cannot answer the question generated from the random goal, thus we want to favour these negative examples in our dataset. We define $\mathbb{P}(\text{keep question}) = \frac{0.325}{1+\exp(6.75-3\text{ number words in common})} + 0.095$. The hyperparameters are chosen empirically to obtain a number of "no_answer" questions similar to the number of questions generated for other words.

**Wide distribution of trajectories**  As explained in the experimental settings and empirically demonstrated in the experiments, a broad distribution of trajectories improves the QA and the efficiency of EAGER overall. To obtain such distribution from the procedural bot, we replace at each time step with a probability $p$ the action of the bot by a random action in: *turn right*, *turn left*, *go forward*, *pick up*, *drop* as it is shown in Figure 2 . For each new example trajectory, we randomly select $p$, with probability 0.45 for $p = 0$ (no random action) and with a probability of $[0.35, 0.1, 0.1]$ for $p \in [0.1, 0.4, 0.8]$ respectively. We use such a distribution to sample $p$ to ensure enough good quality trajectories in the dataset. If the bot cannot complete the task due to the added noise, we discard the trajectory from the set of example trajectories.

We train the QA using the cross entropy loss with a batch size of 10 and a learning rate of $10^{-4}$ using Adam [3]. We multiply the learning rate by 0.1 every 5 epochs. Table 2 summarises the hyperparameters used.

## B.2   Agent architecture and training

We use the actor-critic architecture and the PPO implementation proposed in BabyAI [2] (Figure 3). This PPO implementation uses the default hyperparameters (see Table 3). The output of the actor is the distribution of actions. There are six possible actions: turn left, turn right, forward, pick up, drop, toggle and done (to signify the completion of the task). The output of the critic is the value of the current state.

Goal: Pick up the blue box

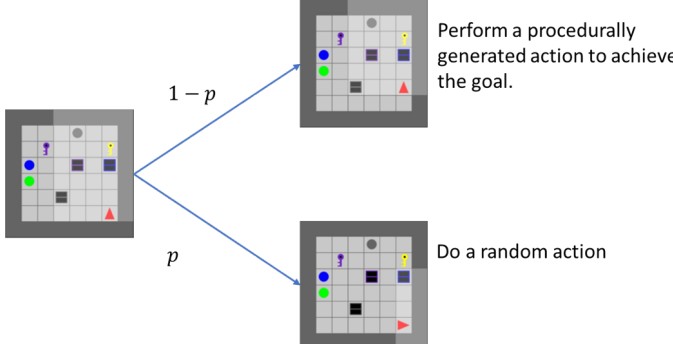

Figure 2: Choice of a new action during the generation of the wide distribution of trajectories. At each time step, the agent can perform a random action with probability $p$.

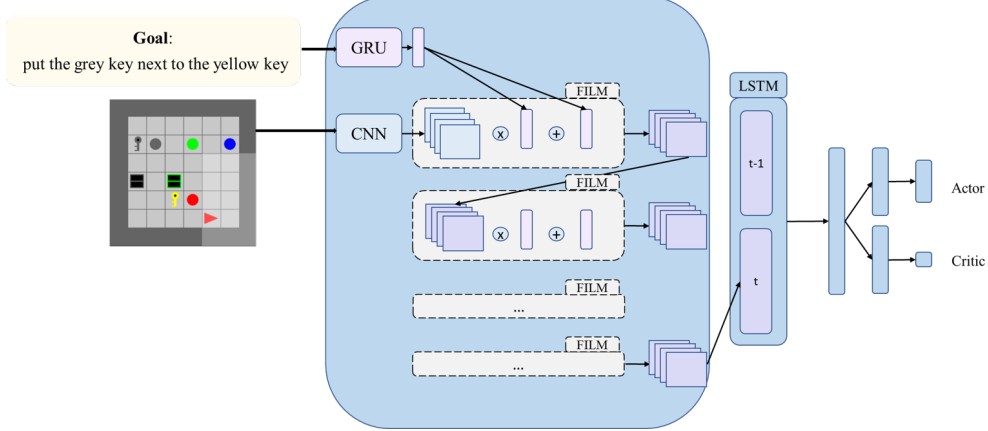

Figure 3: The actor-critic architecture uses a multimodal encoder that mixes language and image using a feature-wise affine transformation FiLM Perez et al. [7]. Then the encoded representation passes through an LSTM to process the history of observations. Finally, the output is used in an actor and critic heads.

## B.3   RIDE

RIDE is a method that does not use language and gives an intrinsic reward encouraging agents to explore actions that significantly alter the state of the environment, as measured in a learned latent space. To use it as a baseline, we reimplemented the RIDE algorithm from the open-source implementation.[1]. We kept the same architecture as the original work and adapted RIDE to the on-policy setting of our PPO algorithm by updating the dynamics models once per batch of on-policy rollouts. We used the hyperparameters values published in the code repository for the coefficients on forward dynamics loss and inverse dynamics loss (10 and 0.1 respectively), as well as the published value for learning rate of $10^{-4}$. We took the intrinsic reward coefficient $\lambda = 0.5$ as it was empirically shown in ELLA to be the best coefficient.

## B.4   ELLA

ELLA is a reward shaping approach for guiding exploration based on the principle of abstraction in language decomposing high-level goals into low-level auxiliary objectives. Two classifiers are learned: a termination classifier that determines when a certain auxiliary objective has ended and a relevance classifier that determines which auxiliary objective is relevant for the high-level goal. For instance, *pick up yellow key* is relevant for the goal *open yellow door*. The termination classifier is trained from labelled trajectories and the relevance classifier is learned online. Expert knowledge is

---

[1]`https://github.com/facebookresearch/impact-driven-exploration`

required to label trajectories and to determine which set of auxiliary objectives is interesting such as: *go to object* or *pick up object*. For this baseline, we rerun the open-source code.[2]

### B.5   EAGER and RIDE

EAGER and RIDE reward different aspects of the exploration, so their combination has the potential to outperform the two methods taken separately. In our experiments we simply add the intrinsic reward returned by EAGER and RIDE, weighted with a shape reward weight $\lambda_{RIDE}$ and $\lambda_{EAGER}$. Nonetheless, we cannot use the technique explained in Section 4 to find the optimal value of $\lambda$. Indeed, this technique is based on the sparsity of the reward, and adding the intrinsic reward from RIDE invalidates this condition. Thus we have to resort to ad hoc methods for tuning $\lambda_{RIDE}$ and $\lambda_{EAGER}$ in the combined version.

We observe that the values of $\lambda$ used by ELLA and EAGER are of the same order of magnitude, so we use the value found in Mirchandani et al. [5] for the combination of ELLA et RIDE. Thus we take $\lambda_{RIDE} = 0.05$ and $\lambda_{EAGER} = 0.1$, we test this combination on PutNext-local, PutNext-medium and Unlock-Medium tasks.

We obtain better results for the task Unlock-Medium Figure 4. For the PutNext-local and PutNext-medium tasks the combination of RIDE and EAGER performs worse. Thus this combination is highly sensitive to the value of $\lambda_{RIDE}$ and $\lambda_{EAGER}$.

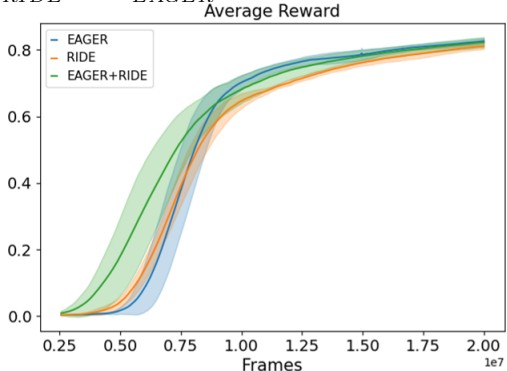

Figure 4: Average Reward for the Unlock-Medium task for EAGER, RIDE and the combination of EAGER and RIDE.

## C   More results on EAGER

**Statistical significance of the results for Unlock-Medium**   For Unlock-Medium the last value for the average return of EAGER is 0.8903, higher than the one of ELLA (0.8765), RIDE (0.8761) and PPO without reshaping (0.8747), see Figure 5. The standard deviation of EAGER is higher (0.011) than the one of ELLA ($6.10^{-4}$), RIDE ($2.10^{-4}$) and PPO without reshaping (0.0021). Indeed, EAGER is still learning and exploring, whereas the baselines are stuck in a local minimum. Moreover, we use a Welch's t-test to test the null hypothesis: equality between the mean of EAGER and ELLA at the end of the curve. The test rejects this hypothesis with $p_{\text{value}} = 1.9\,10^{-31}$. Thus EAGER significantly outperforms ELLA in this task.

**Comparison with behavioural cloning and offline RL**   To train the QA, we used a data set of 7.500 example trajectories per task. One can ask if these trajectories can be used to train an agent through behavioural cloning to obtain results similar to the ones obtained with EAGER. However, experiments performed in [2] show that tasks like PutNextTo-Local require at least 244.000 example trajectories to be learned successfully. Nonetheless, we trained an agent with behavioural cloning using our data set. For the PutNextTo-Local task, our trained agent failed to complete any goal. But when we examine generated trajectories, it seems to display relevant behaviour, going around the objects of interest. This behaviour is coherent with the results given in Mirchandani et al. [5] with 50.000 example trajectories needed for learning goals like *Go to red ball*. Thus EAGER requires much less demonstration trajectories than behavioural cloning, as it is more efficient to train a QA

---

[2]https://github.com/Stanford-ILIAD/ELLA/tree/22884a3da33da2534754693280a47bb0d99eb8c5

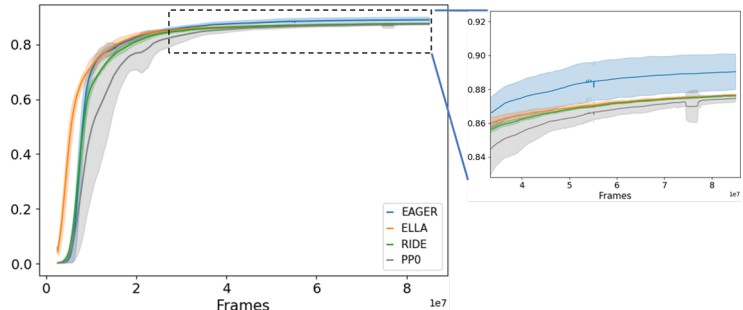

Figure 5: Average Reward for EAGER and baselines.

module than a BC policy. To further this analysis, we have implemented IQL, a recent offline RL algorithm [4]. IQL never needs to evaluate actions outside the data set, but still maintains some generalization capability. Nonetheless, this method also fails due to the small number of trajectories, with a success rate of 0 for PutNextTo-local tasks. To ensure that the implementation is correct, we tested IQL with a simple GoTo task. It obtains a non zero result on the test set with $6\%$ of the test trajectories being successful. Thus the data set used for training the QA cannot be used to warm start an agent. This result is not too surprising, it is indeed easier to learn from a data set to recognise a pattern (as the QA module does) rather than learning a policy Bahdanau et al. [1].

**Generalisation properties of the QA module**  We also verified that the QA can generalise by correctly answering questions formulated from goals never seen in training. To check this, we trained the QA with questions and trajectories generated from the Sequence-Medium task. To test generalisation, we only used in the test data set trajectories and questions generated from goals that are not present in the train data set. On the test set, we obtained a success rate of $0.67$ to be compared with $0.74$ when we tested with trajectories with already seen goals. Thus the QA module can generalise to new goals, making EAGER efficient even for goals not present in the QA training data set.

**Ablation of the policy invariance**  To ensure the policy invariance requirement (see Section **??**) we subtracted the shaped reward at the final step of successful trajectories by neutralizing the intrinsic reward. In Figure 6, the curve in orange is the learning curve when we do not apply neutralization. The agent learns faster that the one on which neutralization is applied because it receives more indication through reward. However, it ends up stuck in a local minimum because the final policy is influenced by the intrinsic reward.

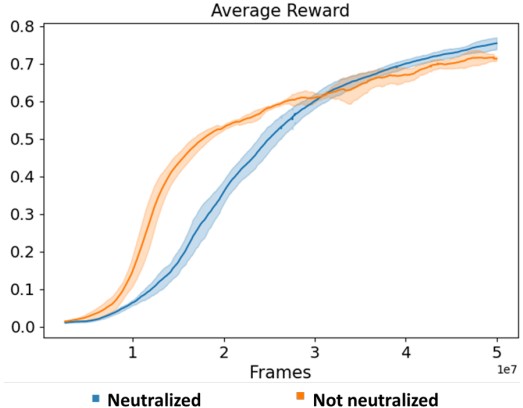

Figure 6: Ablation of the neutralization of the intrisic reward on the task PutNextTo-local.

**QA performance along the trajectory**  In order to minimise human intervention, we only use whole trajectories to train the QA. It is legitimate to ask whether the performance of the QA module changes along the trajectory. To determine this, we used the number of different attempts before

successfully answering a question as a proxy to measure the performance of the QA module on short trajectories.

We used the PutNext-local task which has the following structure: "Put the adjective_1 noun_1 next to the adjective_2 noun_2". Measuring the number of different attempts before finding the correct answer, we began to count after the first answer different from "no_answer" (before that, the QA estimates that it cannot answer). For this task, after 100 trajectories, we found:

| Token | Average number of attempts |
|---|---|
| adjective_1 | 4.37 |
| noun_1 | 2.55 |
| adjective_2 | 1.69 |
| noun_2 | 1.25 |

It appears that the QA module needs on average twice as many attempts to guess the answer at the beginning (when the trajectory is short and partial) rather than at the end of the trajectory. Nonetheless, in practice, this does not seems to impact the performance of the methods, which underlines the robustness of EAGER.

**Results for the task Sequence with an extended time budget**    In order to understand why EAGER suddenly does better than ELLA after 6.5e8 frames Figure 7, we empirically observed which tasks were successful before and after this threshold. The goals in Sequence task are a combination of two tasks among 'Pickup', 'Go to','Open', and 'Put Next to'. Nonetheless, the later is already a combination of the tasks 'Pickup' and 'Go to', thus more difficult than any of them alone. In Figure 8 we observe the reward obtains by ELLA and EAGER for goals without 'Put Next to' tasks in it, with 1 'Put Next to' and 2 'Put Next to'. The former type represents around $50\%$ of the goals, goals with 1 'Put Next to' represents $40\%$ and the latter accounts for $10\%$ of the goals. Thus, after 6.5e8 frames, the improvement of the EAGER agent for the goals with 1 and 2 'Put Next to' allows for an important increased in the total reward.

Up to 6.5e8 frames EAGER and ELLA only mostly succeed on Sequence tasks that does not contains a 'Put Next to' goal. However, when the agent starts to have a trajectory that brings it closer to success for tasks with 'Put Next to' in them, it seems that EAGER recognises it and rewards it. In contrast ELLA is much slower as it must first learn to break down complex goals (with one or two 'Put Next to' tasks in them) into auxiliary objectives (using its relevance classifier) before it can use them for reward shaping.

If we look at the curves of ELLA and EAGER we can try to interpret all the different phases:

- At the beginning (0 to $1, 5.10^8$ frames) ELLA does better than EAGER because the relevance classifier breaks down the goal into auxiliary objectives quite easily, whereas EAGER is hampered by the fact that the trajectories are shorter and performs poorly (as shown in the paragraph: *QA performance along the trajectory*)
- After ($1, 5.10^8$ to $3, 5.10^8$) the agent's trajectories improve and become longer, EAGER gains in performance and overtakes ELLA
- From $3, 5.10^8$ to $6, 5.10^8$ EAGER and ELLA manage to achieve almost the same goals (goals with no 'Put Next to tasks in it)
- From $6, 5.10^8$ frames the agent's trajectories for complex goals are longer, EAGER is performing well while ELLA is training the relevance classifier to decompose complex goals into auxiliary objectives (this is learned online from the agent's successful trajectories which explains the slow performance)

**Verifying that the QA does not guess**    With the QA, the agent can self-check the followed instructions. Due to the format of the generated questions (i.e. masking a word) and proceeding by elimination depending on the object present in the environment, a simple QA could guess the answer, breaking the EAGER method. Thus, to avoid such issues we added so-called "no_answer questions" to the QA training data set. These questions are examples where the trajectory does not correspond to the question, e.g. a trajectory corresponding to the goal "pick up the red box then pick up the red

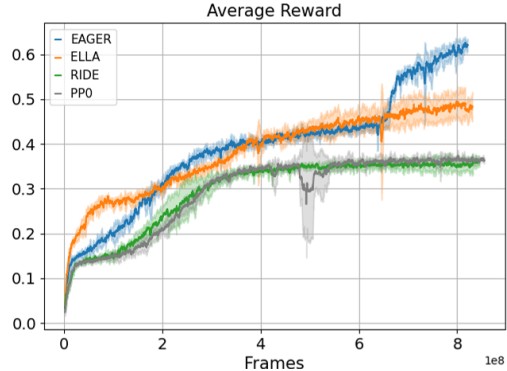

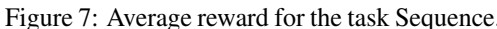

Figure 7: Average reward for the task Sequence.

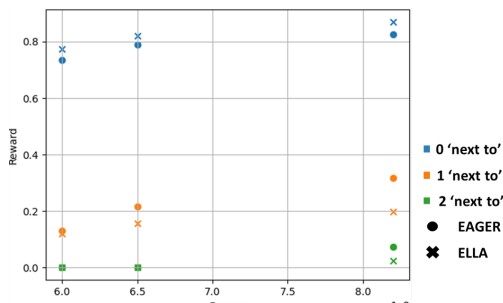

Figure 8: Average reward over 500 goals and 4 seeds, in Sequence, distributed in three categories: goals without 'Put next to' task in it and goals with 1 or 2 'Put next to' tasks.

ball" is associated to the question "pick up the «question» key then pick up the red ball". The QA must learn to associate each element to see that the path is not associated to the correct question and appropriately answer "no_answer". In practice, this prevents the QA from guessing the answer, it needs to wait for the agent's path to match before answering. Thus for a question such as "pick up the «question» box then pick up the red ball", even if the only box in the room is red, the QA must answer "no_answer" as long as the agent has not completed the corresponding auxiliary objective.

We verified that the QA correctly acts by checking a hundred trajectories with the possibility to guess an answer based on the linguistic input, such as with the goal "pick up the red ball then pick up the red ball". We give 4 examples among the tasks we have checked in Figure 9. We did not find any instance where the QA guessed the answer by chance right from the beginning of a trajectory. Indeed, even if it is possible to just use linguistic elements to answer a question such as "pick up the «question» ball then pick up the red ball", the QA cannot associate the answer "red" to any element of the trajectory and thus returns "no_answer".

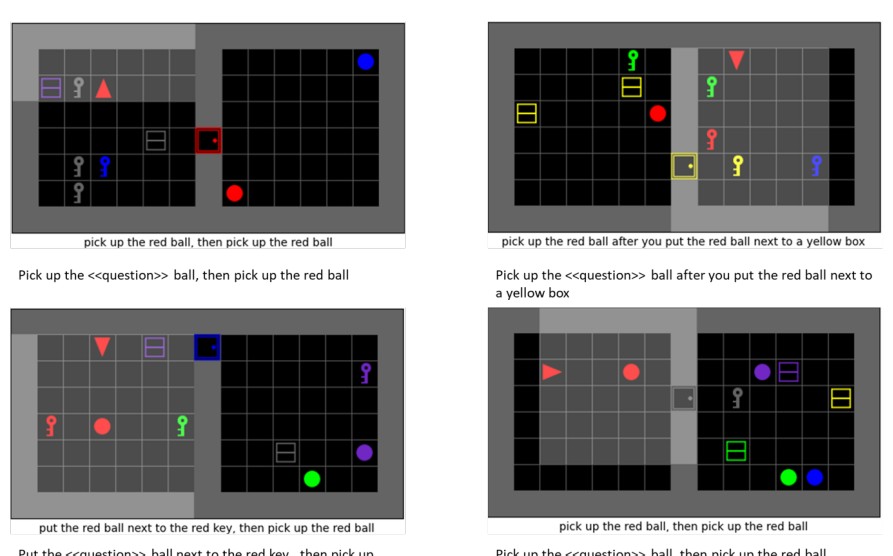

Figure 9: Four examples and possible question at time step $t = 0$ where the QA could guess the answer either linguistically or by elimination process. In all of these examples the QA returns no_answer because the trajectory does not correspond to a possible answer

**Relation between sample efficiency and success rate of the QA**     In the experiments on EAGER robustness, we looked at the robustness of EAGER relatively to the success rate of the QA during pre-training. However, we only used some values of SR — by sampling the QA at different training

epochs— to train the RL agent. To strengthen our point, we used a Gaussian Process model Rasmussen and Williams [8] to fit the relation between the success rate of the QA and the sample efficiency (SE) of EAGER. The SE is defined as $SE = \frac{1}{n_{frames}} \sum_{i}^{n_{frames}} r_i$), where $r_i$ is the extrinsic reward and $n_{frames}$ the number of frames seen during training. Figure 10 shows once again that EAGER is robust to the quality of the QA module with an almost constant sample efficiency when $SR > 0.52$

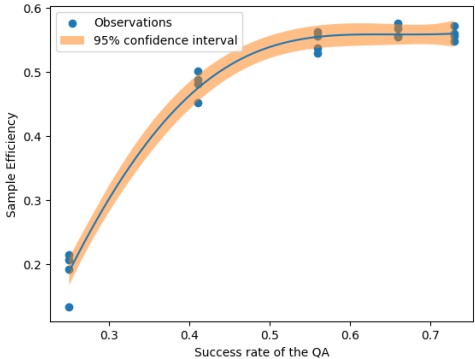

Figure 10: Gaussian process model to fit Sample Efficiency (SE) of the EAGER algorithm function of Success Rate (SR) of the QA. For each value of SR, the different points correspond to the different seeds used to train the RL agent on the PutNextTo task.

**EAGER, a precise guiding technique**    To understand why EAGER achieves better results than ELLA or RIDE, we propose an answer based on qualitative observations. We believe that EAGER guides the agent more finely. The generated questions break down the trajectory into more precise key points than the auxiliary objectives generated in ELLA. To verify our point, we plot the average trajectory over 4 seeds for agents trained with EAGER or ELLA, see Figure 11. Looking at the average trajectories, it appears that EAGER generates trajectories that are spatially more compact and that return to the same place fewer times.

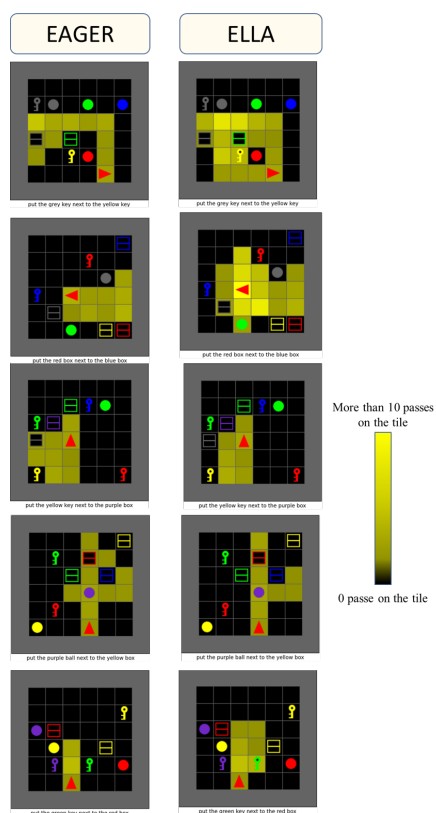

Figure 11: Average trajectory for PutNextTo-Local tasks for EAGER and ELLA. The colour of the tiles depends on the average number of passes of the agent on the tile. The 5 environments presented here have been randomly selected. The position of the agent is its position at the beginning of the episode.

# D   Commented EAGER algorithm

Algorithm 1 outlines the pseudo-code of our learning architecture. At the beginning of an episode, the EAGER algorithm uses the goal to generate an active set of questions $\mathcal{Q}$ that is updated at each time step by removing correctly answered questions. Each question can be understood as an auxiliary objective guiding the agent towards the completion of the main goal. When an auxiliary objective is completed (i.e. the corresponding question is correctly answered), the agent receives an intrinsic reward $\lambda$. To avoid modifying the optimal policy with these additional rewards, we substract the sum of discounted intrinsic rewards from the reward at the last step of successful trajectories with the NEUTRALISE function. In ELLA [5], the authors prove that shaping the reward does not change the optimal policy when using this neutralisation procedure.

---

**Algorithm 1** Automatic auxiliary goal generation and reward shaping using EAGER

---

**Input:** $\theta_0$ initial policy parameters, $\lambda$ bonus scaling factor, ENV the environment and OPTIMISE an RL optimiser

    **for** k=0,..., $n_{step}$ **do**
        $\omega^g, o_0, \text{done}_0 \leftarrow$ ENV.$reset()$
        $\mathcal{Q}_0 = \{q_1, ..., q_k\} \leftarrow QG_k(\omega^g)$                           $\triangleright$ Generate questions by masking words
        $t \leftarrow 0$
        **while** $\text{done}_t$ not True **do**
            $a_t \leftarrow \pi^{\theta_0}(o_t)$
            $o_{t+1}, r_t, \text{done}_{t+1} \leftarrow$ ENV$(a_t)$
            $r'_t, \mathcal{Q}_{t+1} \leftarrow$ QA_SHAPE$(\mathcal{Q}_t, \tau, r_t)$    $\triangleright$ Shape reward and update active set of question $\mathcal{Q}$
            **if** $\text{done}_{t+1}$ is True **then**
                $N \leftarrow t$
                $r'_N \leftarrow$ NEUTRALISE$(r'_{1:N})$
            **end if**
        **end while**
        Update $\theta_{k+1} \leftarrow$ OPTIMISE$(r'_{1:N})$
    **end for**

    **function** QA_SHAPE$(\mathcal{Q}_t, \tau, r_t)$
        **for** $q$ in $\mathcal{Q}_t$ **do**
            $\tilde{a}^* \leftarrow QA(q, \tau)$                      $\triangleright$ Answer $q$ using the trajectory, $\tau = (o_i, a_i)_{i \in [|0,t|]}$
            **if** $\tilde{a}^*$ is correct answer to q **then**
                $r'_t = r_t + \lambda\, QA(\tilde{a}^* \mid q, \tau)$
                $\mathcal{Q} = \mathcal{Q} \setminus \{q\}$                      $\triangleright$ Update the active set of questions
            **end if**
        **end for**
        **return** $r'_t, \mathcal{Q}$
    **end function**

    **function** NEUTRALISE$(r'_{1:N})$
        $r'_N \leftarrow r'_N - \sum_{t \in T_{\mathbb{S}}} \gamma^{t-N} r'_t$      $\triangleright$ $T_{\mathbb{S}}$ time steps where the agent has received a shape reward
        **return** $r'_N$
    **end function**

---

# E   Hyperparameters tables

This section contains three tables: the hyperparameters used for training the QA in Table 2, the hyperparameters for the PPO algorithm in Table 3, and the shape reward value $\lambda$ depending on the task in Table 4.

Table 2: QA training hyperparameters

| Variable | Value |
|---|---|
| batch size | 10 |
| learning rate (lr) at the beginning | $10^{-4}$ |
| number of steps before decreasing lr | 5 |
| factor decrease | 0.1 |

In Table 4 we give all the elements we use to compute $\lambda$, which is the value of the intrinsic reward: $\lambda = \frac{\gamma^N r_N}{k}$, where $\gamma = 0.99$ is the discount factor. $r_t = 20(1 - 0.9\frac{t}{H})$ is the reward obtained for completing the goal at step $t$, with $H$ the maximum number of steps for a given task. In the calculation of $\lambda$, we assume that once trained, the agent completes the goal in $N$ steps in the worst case scenario. $k$ represents the maximum number of questions that can be generated from goals of a certain task.

Table 3: PPO hyperparameters

| Variable | Value |
|---|---|
| batch size | 2560 |
| mini-batch size | 1280 |
| discount factor | 0.99 |
| lr | $7 \times 10^{-4}$ |
| entropy coefficient | 0.01 |
| loss coefficient | 0.5 |
| clipping-$\epsilon$ | 0.2 |
| generalised advantage estimation parameter | 0.99 |

Table 4: Value of $\lambda$ depending on the task.

| Task | k | H | N | $\lambda$ |
|---|---|---|---|---|
| PutNextTo-Local | 4 | 128 | 40 | 2.4 |
| PutNextTo-Medium | 4 | 256 | 80 | 1.6 |
| Unlock-Medium | 2 | 128 | 40 | 4.8 |
| Sequence-Medium | 9 | 512 | 185 | 0.23 |