# OpenReview forum: "EAGER: Asking and Answering Questions for Automatic Reward Shaping in Language-guided RL"
_NeurIPS.cc/2022/Conference — NeurIPS 2022 Accept_

### Official Review · Reviewer_aQK3 · 2022-06-24

**Rating:** 4
**Confidence:** 4
**Soundness:** 3 good
**Presentation:** 3 good
**Contribution:** 3 good

**Summary:**

In this paper, the authors proposed a reward shaping method for language-conditioned RL. The authors proposed to generate a set of questions based on natural language instructions, and then evaluate how likely the current trajectory answers the questions. The probabilities of the questions being correctly answered are used as intrinsic rewards. In the experiment, the authors worked on Baby AI benchmark and showed that the proposed reward shaping method helped to improve sample complexity and model performance.

**Questions:**

Section 4.1 mentioned that if a question is answered, then it is removed from the set of questions. How to determine if a question is answered? Does the QA model have to predict the correct answer with high probability?

**Strengths And Weaknesses:**

Strengths:
The idea of using question generation and answering to do reward shaping for LC-RL is novel and interesting. It is like doing a self-check when a human follows a task instruction.

The paper showed that this idea works in the Baby AI benchmark.

Weaknesses:
While the idea is interesting, I think it is not sophisticated enough to generalized to more complex setting.

First of all, the QG module generates a questions by masking a word (such as nouns and adjectives) in the linguistic instruction. However, for complex instructions, it is likely that the question can by answered by the masked instruction. For example, an instruction "pick the [red ball], open the red door, ...., put the [red ball] on the chair". If we mask the 'red ball' at the beginning of the sentence, we can still infer the answer from the 'red ball' at the end of the sentence. Therefore, even without the trajectory, the questions can be answered. I'd assume that it requires more than just masking words to generate a valid question.

Secondly, the QA model is trained by using the full trajectory, however, when calculating intrinsic rewards, partial trajectories (trajectories up to the current timestep) are used. I would assume that when the partial trajectories are short, the QA model performance is bad. Have the authors investigated this?

---

> ### Author Response · Authors · 2022-08-02
> **Response to the reviewer aQK3: clarification about the QA module**
>
> We thank reviewer aQK3 for the constructive feedback and hope to clarify our method through this discussion. We first explain how we prevent the QA module from guessing an answer, then give more detail on the QA performance relative to the length of the trajectory, and finally how to determine if a question can be removed from the set of questions.
>
> ### 1) Preventing the QA from guessing
>
> The reviewer is concerned about the possibility of the QA guessing the answers without using the path.
>
> As Reviewer aQK3 noted, with the QA the agent can self-check the instructions followed. Due to the format of the generated question (i.e. masking a word) and proceeding by elimination depending on the object present in the environment, a simple QA could guess the answer, breaking the EAGER method. Thus, to avoid such issues we added so-called “no_answer questions” to the QA training dataset. These questions correspond to examples where the trajectory does not correspond to the question, e.g. a trajectory corresponding to the goal "pick up the red box then pick up the red ball” is associated with the question "pick up the <<question>> key then pick up the red ball". The QA must learn to associate each element to see that the path is not associated with the correct question and appropriately answer “no_answer”. In practice, this prevents the QA from guessing the answer: it needs to wait for the agent's path to match before answering.
>
> Thus for a question such as  "pick up the <<question>> box then pick up the red ball”, even if the only box in the room is red, the QA must answer “no_answer” as long as the agent has not completed the corresponding auxiliary objective.
>
> We have verified that the QA correctly acts by checking a hundred trajectories with the possibility to guess an answer based on the linguistic inputs, such as with the goal “pick up the red ball then pick up the red ball”. We did not find any instance where the QA guessed the answer by chance right from the beginning of a trajectory. Indeed, even if it is possible to just use linguistic elements to answer a question such as  “pick up the <<question>> ball then pick up the red ball”, the QA cannot associate the answer “red” to any element of the trajectory and thus return “no_answer”.
>
> ### 2) QA performance along the trajectory
>
> The reviewer wants to know how the performance of the QA module changes along the trajectory.
>
> In order to minimize human intervention, we only use whole trajectories to train the QA. It is legitimate to ask whether the performance of the module changes along the trajectory. To determine this, we used the number of different attempts before successfully answering a question as a proxy to measure the performance of the QA module on short trajectories. We used the PutNext-local task which has the following structure: “Put the adjectif_1 noun_1 next to the adjectif_2 noun_2”. Measuring the number of different attempts before finding the correct answer, we began our count after the first answer different from “no_answer” (before that, the QA estimates that it cannot answer). For this task, after 100 trajectories,  we found:
>
> - adjective_1: 4.37  attempts on average before guessing correctly
> - noun_1: 2.55 attempts
> - adjective_2: 1.69  attempts
> - noun_2: 1.25 attempts
>
> So the QA module tends to perform worse at the beginning of the trajectory.
> It appears that the QA module needs on average twice as many attempts to guess the answer at the beginning (when the trajectory is short and partial) rather than at the end of the trajectory. Nonetheless, this does not seem to impact the global performance of the method in practice, so this also highlights the robustness of EAGER.
>
> ### 3) Removing a question
>
> The reviewer wants to know how to determine if a question is correctly answered.
>
> The advantage of the masking process used by the QG is that the answer is known. A question is only removed from the set of questions if the given answer matches the masked word. EAGER is robust, precisely because it ensures that the agent can verify the correct answer, and does not require the complexity of having a mechanism to measure uncertainty. In more complex settings where the QG is not just a masking module, a sensible way to remove a question would be to remove it if the QA predicts the answer with a sufficiently high probability.

---

> > ### Author Response · Authors · 2022-08-08
> > **Follow-up to Reviewer aQK3**
> >
> > Thank you again for your review. We wanted to quickly follow up and see if our response adequately addresses your questions and comments, or if you have additional concerns.

---

> > > ### Comment · Reviewer_aQK3 · 2022-08-09
> > > **Thanks for the responses from the authors.**
> > >
> > > For the discussion regarding guessing the answers: I am not convinced that by adding a `no answer' option would resolve this issue. The statement from the author response is not backed by numbers. The paper should include an experiment which trains a QA system without conditioning on the trajectories, and then evaluate the performance on questions that has answers and see how much the QA performance regresses.
> > >
> > > For the discussion regarding partial trajectories. From the authors response, the QA indeed performs worse at the beginning of the trajectories. However, as long as this statement is included in the paper for clarification, I think it is not a concern given the other contribution of this paper.

---

### Official Review · Reviewer_ZfBg · 2022-07-12

**Rating:** 8
**Confidence:** 4
**Soundness:** 4 excellent
**Presentation:** 4 excellent
**Contribution:** 4 excellent

**Summary:**

The paper proposes an auxiliary reward scheme (EAGER) for instruction following RL (BabyAI framework) based on question generation (QG) and answering (QA). Concretely, QG is based on masked word modeling of the instruction, and QA is pre-trained with demo instructions and trajectories and fixed when training RL. The intuition is good and successful RL agent trajectories should make QA correct and easy, thus when QA is successful, an auxiliary reward proportional to correct answer confidence is given. The method is evaluated on several BabyAI tasks and shown on par with other methods (ELLA, RIDE) with more expert inductive bias for subgoal specification and judgement. Further analysis shows EAGER is fairly robust to QA performance, but some QA design choices are important.

**Questions:**


Some minor points:

- Maybe in abstract, mention the domain (babyAI)?
- Is there any ablation for the policy invariance thing?

**Limitations:**

They are discussed in the paper, see the weakness part.

**Strengths And Weaknesses:**

Strengths:

- The idea is neat, novel, effective, and provides a means to auxiliary reward for instruction following only using example instructions and trajectories, and minimal linguistic assumptions (what is noun/adj). In contrast, previous methods usually require domain knowledge about subgoal formulation/proposal/completion judgement.

- Some design choices for QA are reasonably explained in theory (e.g. policy invariance) and practice (e.g. no_answer, using confidence instead of binary reward).

- The motivation and idea are well explained, and the paper is well structured and easy to read for most parts.


Weakness:

As mentioned in the last part of the paper, two main limitations (and some of my extensions) are

- QA needs expert trajectories with instructions to pre-train. Arguably it is a weaker assumption compared to prior work, but I feel two important ablations are missing: direct PPO without the auxiliary reward (a trivial low-bound baseline), and just using the QA training trajectories to do imitation learning and warmstarting for RL agent (a trivial way to use these pre-training trajectories).

- BabyAI is perhaps too simple, and experiments seem a bit limited. I can see some sample efficiency advantage on PutNext tasks, but ELLA seems better than EAGER on Unlock and Sequence. What about other tasks in BabyAI, let alone more complex tasks like Alfred (which I feel this method might not easily work, given the visual observation and language instruction are both much harder). Some more experiments or tasks would make me believe the method is more general and robust.

Also pointed out in the paper (line 295),'

- EAGER might not work well on Sequence or tasks with temporal subgoals. In general, I don't have an intuition what properties of env/task does EAGER work better on and what not. Some analysis or examples of agent behavior or even how auxiliary reward changes with time would give me more concrete ideas.

---

> ### Author Response · Authors · 2022-08-02
> **Response to reviewer ZfBg: clarification about EAGER capacity**
>
> We are grateful for the positive comments of the reviewer ZfBg underlying the novelty and effectiveness of our method. In order to respond to the various points raised by the reviewer, we first detail our experiment with behavioural cloning then we give more information about the capacity of EAGER.
>
> First, as suggested by the reviewer, the baseline PPO without reward shaping has been added to the paper. As expected, this baseline performs much worse than the other two. We thank the reviewer for the comment about an ablation of the policy invariance. The ablation has shown the importance of this aspect and the result has been added to the supplementary in Ablation of the policy invariance (Part C).
>
> ### 1) Comparison with behavioural cloning and offline RL
>
> The reviewer wonders whether the trajectories that trained the QA could be used to train the agent with BC
>
> In order to check whether the trajectories generated for training the QA can be used to train an agent using behavioural cloning (BC), we performed a detailed experiment in the supplementary material. We trained an agent with BC on our dataset. However, as expected in regards to the findings of the BabyAI paper [1], 7.500 noisy trajectories are not enough to learn a task (we obtained a success rate of 0 on all the tasks). Even the simplest one like PutNext requires theoretically at least 244.000 example trajectories [1]. We also tried offline RL with the recent IQL algorithm [2], nonetheless, even using IQL the agent fails to learn a simple task like PutNextTo-local.  We had to design a straightforward GoTo task where this method obtains a non-zero score on the test set (it gets 6%) to make sure that all the zero scores were not due to a bug.
>
> ### 2) EAGER capacity
>
> The reviewer asks for more precision concerning EAGER capacities.
>
> To test EAGER, we performed experiments on 4 tasks that illustrate the better capacity of EAGER under sparsity and temporal constraints. But so far, our paper is just a proof of concept for a new reward shaping technique in LC-RL.  In future work, we are willing to extend this study to more complex environments such as ALFRED.
>
> Finally, with an extended time budget for training, we obtained new results showing that EAGER outperforms ELLA and other baselines in all the tasks. Changes have been made in the paper to reflect these new observations.
>
> As reviewer aQK3 pointed out, we only used whole trajectories to train the QA. It is legitimate to ask whether the performance of the module changes along the trajectory. We have added an experiment in the supplementary QA performance along the trajectory (Part C) where we show that performances of the QA are indeed poorer at the beginning of the trajectory. Nonetheless, this does not seem to impact the global performance of the method in practice, highlighting the robustness of EAGER. This helps us better understand why EAGER is less effective in early learning of the Sequence task. Indeed, this task contains long linguistic instructions which make EAGER less efficient and less useful at the beginning of the trajectory. However, as the agent improves and completes a longer part of the linguistic instructions, EAGER becomes more and more efficient, explaining why EAGER finally outperforms ELLA.
>
> [1] Maxime Chevalier-Boisvert, Dzmitry Bahdanau, Salem Lahlou, Lucas Willems, Chitwan Saharia, Thien Huu Nguyen, and Yoshua Bengio. Babyai: A platform to study the sample efficiency of grounded language learning. In International Conference on Learning Representations (ICLR), 2019.
>
> [2] Ilya Kostrikov, Ashvin Nair, Sergey Levine Offline Reinforcement Learning with Implicit Q-Learning, 2021

---

> > ### Comment · Reviewer_ZfBg · 2022-08-07
> > **Thank you**
> >
> > Thank the authors for addressing my questions, and I believe the new experiments have made the paper stronger. I would raise my score from 7 to 8 to show my support, and encourage fellow reviewers to engage in discussions and see if questions have been addressed.

---

> > > ### Author Response · Authors · 2022-08-08
> > > **Follow-up to Reviewer ZfBg**
> > >
> > > We are pleased to hear that our response has addressed your questions and concerns. Thank you again for taking the time to respond.

---

### Official Review · Reviewer_JBKz · 2022-07-14

**Rating:** 6
**Confidence:** 4
**Soundness:** 3 good
**Presentation:** 3 good
**Contribution:** 3 good

**Summary:**

This paper presents a simple but novel reward shaping method for language-conditioned RL. The paper mainly proposes the auxiliary objectives for question generation and question answering models. Then they calculate the intrinsic reward which is proportional to its confidence in its answer. In the experiments, the authors showed that the proposed algorithm learns more efficiently on various tasks than the baseline algorithms.


**Questions:**

- In lines 280-281, it is expressed as without using expert knowledge, but is the expert dataset not used for pre-training when learning the QA module? (The paper explains as if it used less information than the ELLA, but it requires the same or similar information as the ELLA.)
- If there is a pre-trained dataset (for the QA module), isn't it more efficient to use knowledge in the form of a prior policy learned from it? It seems that there should be an experiment about it.
- What if it is used simultaneously with intrinsic reward methods such as RIDE? It is also important to check whether the effect of both methods is still helpful even if they are used at the same time.


**Limitations:**

The authors adequately addressed the limitations and potential negative societal impact of their work.


**Strengths And Weaknesses:**

Strengths
- The main idea of the proposed method in the paper is simple but novel.
- The experiments include not only the results of sample efficient learning but also various analyzes.

Weaknesses
- A near expert dataset is needed for pre-training the QA module, which seems to be a quite strong assumption.

---

> ### Author Response · Authors · 2022-08-02
> **Response to reviewer JBKz: clarification on the use of expert knowledge**
>
> We thank the reviewer for their positive comments on the novelty of our method and the thoroughness of our analyses. In the following, we address the concerns the reviewer has pointed out regarding the dataset used to train the QA and the possibility to use behavioural cloning (BC).
>
> ### 1) Dataset and expert knowledge
>
> The reviewer finds that the use of a data set to train the QA undermines the claim about the low use of expert knowledge.
>
> In order to clarify our argument about the low use of "expert knowledge" by EAGER with respect to ELLA, we would like to emphasize two points:
>
> First and foremost, ELLA requires human experts that have to determine which class of auxiliary objective has to be associated with the task. The point of EAGER is to remove this human intervention as EAGER is able to extract auxiliary objectives and determine their completion coupled with the associated intrinsic reward through the QG and QA modules. As the reviewer ZfbG points out, our method does not require “domain knowledge about subgoal formulation/proposal/completion judgment”. The only human intervention is about how to determine what words are used for the question, but it merely consists in masking nouns and adjectives, and could be easily automated.
>
> Moreover, even if both ELLA and  EAGER use demonstrations, their quality differs. By contrast with ELLA, the trajectories used in EAGER are noisy, underlying its robustness. Indeed, as explained in the paragraph Wide distribution of trajectories in the supplementary (Part C), the bot used in EAGER can take a random action with probability p at each step. Thus, trajectories used in EAGER are neither optimal nor efficient, they only successfully complete the task possibly with very suboptimal trajectories.
>
>
> ### 2) Comparison with behavioural cloning and offline RL
>
> The reviewer asks if it could be interesting to warm start the agent using behavioural cloning on the data used for training the QA.
> As underlined by the reviewer, it is important to verify if the dataset of generated trajectories is complete enough to train a policy. We have conducted experiments presented in the supplementary material (line 94 Comparison to behavioural cloning) to see if we can obtain a relevant policy using the example trajectories given to the QA. However, as the BabyAI paper [1] has shown, 7.500 noisy trajectories are by far not enough to learn a policy. Even simple tasks like PutNext require at least 244.000 training trajectories to be successfully learned. To complete and deepen this analysis, we have implemented IQL, a recent  offline RL algorithm [2]. Nonetheless, this method also completely fails due to the small number of trajectories. We had to design a straightforward GoTo task where this method obtains a non-zero score on the test set (it gets 6%) to make sure all the zero scores were not due to a bug.
>
> ### 3) Combining EAGER and RIDE
>
> The reviewer suggests combining EAGER and RIDE. We have tried to follow the suggestion. However, due to the short time available, we were not able to test different hyperparameter combinations. Thus we use the one given in ELLA and obtain conclusive results only for the Unlock-Medium task. We added these results to the supplementary. More research to obtain an efficient combination of both methods is left for future work.
>
> [1] Maxime Chevalier-Boisvert, Dzmitry Bahdanau, Salem Lahlou, Lucas Willems, Chitwan Saharia, Thien Huu Nguyen, and Yoshua Bengio. Babyai: A platform to study the sample efficiency of grounded language learning. In International Conference on Learning Representations (ICLR), 2019.
>
> [2]Ilya Kostrikov, Ashvin Nair, Sergey Levine Offline Reinforcement Learning with Implicit Q-Learning, 2021

---

> > ### Author Response · Authors · 2022-08-08
> > **Follow-up to Reviewer JBKz**
> >
> > Thank you again for your review. We wanted to quickly follow up and see if our response appropriately addresses your questions and comments, or if you have any further concerns.

---

> > > ### Comment · Reviewer_JBKz · 2022-08-09
> > > **Response to author response**
> > >
> > > Thank you for providing the rebuttal to respond to my questions. Most of my questions have been addressed. I will keep my score unchanged.

---

> > > > ### Author Response · Authors · 2022-08-09
> > > > **Follow-up to Reviewer JBKz**
> > > >
> > > > We are pleased to hear that our response has addressed most of your questions and concerns. Thank you again for taking the time to respond.

---

### Official Review · Reviewer_piZu · 2022-07-16

**Rating:** 6
**Confidence:** 3
**Soundness:** 3 good
**Presentation:** 3 good
**Contribution:** 3 good

**Summary:**

This paper proposes a new reward shaping method, EAGER, in Language-guided reinforcement learning (RL).  By extracting auxiliary Question Answering (QA) tasks, the agent is given intrinsic rewards based on how the generated trajectories could help answer these questions.  The authors claim that such auxiliary tasks can be designed without engineer interventions.  Specifically, the answer to these questions is the output of the pre-trained QA transformer model after feeding the generated trajectories.  For the BabyAI experiments, this QA model is pre-trained using example trajectories provided by the bot that can already solve the environment. The method is compared against two recent baselines RIDE and ELLA.  EAGER improves sample-efficiency in some tasks from BabyAI.

**Questions:**

- Is it possible to use behavior cloning to train an agent using those example trajectories from the bot?  This might serve a good baseline.


**Limitations:**

Yes, the authors have addressed the limitations

**Strengths And Weaknesses:**

Strengths:
-This seems like a novel method that is inspired by reference-less metrics in NLP. The idea is interesting, and the authors provided detailed explanations on how each component, QA module, QA module, is implemented.
- Overall, the paper is well-structured and clearly written from the method section to the experiment section.
- I believe the future work could be influenced by the idea of reference-less metrics proposed here.

Weaknesses:
- I think the experiment part is weak due to EAGER still using “engineer intervention” in BabyAI.  If I understand correctly, RIDE [1] does not use any expert information at all while EAGER currently requires a bot that can already solve some of the tasks from BabyAI.  In addition,  QG assumes the answer to the generated questions is one of the words that appear in the goal sentence.  The output of the QA model also requires prior knowledge about the potential words that can appear in the goal sentence though this might be okay if you assume these  are all nouns and you could extract them in advance.  I recommend adding a table to list all the assumptions and expert knowledge required for RIDE, EAGER, and ELLA [2] in a table.
- Although EAGER outperforms RIDE, RIDE does not use the bot from BabyAI so I think it’s not too surprising that EAGER can outperform RIDE all the time.  Since EAGER and ELLA both excel in certain tasks, it is difficult to judge whether the method is actually an improvement to ELLA given the reasons in the previous bullet point.

Minor points:
- Title 5.2 “How does EAGER performs” → “How does EAGER perform” “Sparsity increase” → increases
- Line245: I think it’ll be helpful to explain what these 3 input values represent in BabyAI.  How are objects represented in the gridworld.


[1] Roberta Raileanu and Tim Rocktäschel. Ride: Rewarding impact-driven exploration for 418 procedurally-generated environments. In International Conference on Learning Representations 419 (ICLR), 2020

[2] Suvir Mirchandani, Siddharth Karamcheti, and Dorsa Sadigh. Ella: Exploration through learned 406 language abstraction. In Advances in Neural Information Processing Systems (NeurIPS), 2021.

---

> ### Author Response · Authors · 2022-08-02
> **Response to reviewer piZu: clarification on the comparison with ELLA and the use of a dataset (1/2)**
>
> We warmly thank the reviewer for their positive assessment of the clarity of our work and the potential impact of our reference-less metric approach on future work, as well as for useful and constructive suggestions. We have corrected all the minor mistakes pointed out in the revised version of the paper. Below we address the weaknesses spotted by the reviewer as well as the other minor points.
>
> First, as suggested by the reviewer, we have added a table listing all the assumptions and expert knowledge required for RIDE, EAGER, and ELLA, see Section A of the revised supplementary material. We agree that such a table clarifies our point.
>
> ### 1) Comparison to ELLA
>
> The reviewer considers that EAGER does not use less expert knowledge than ELLA and thus that it is difficult to see if there is some improvement between these two reward shaping methods.
>
> We compare EAGER to a previous method also using high-level descriptions with language for reward shaping: ELLA. We claim that EAGER uses less “engineer expertise” than ELLA while still getting better results. Our point is twofold:
>
> The first and most important point where we lessen the need for “engineer expertise” is the decomposition of the goal into auxiliary objectives. In ELLA, human experts decide for each task what auxiliary objective has to be used (for instance PutNext is associated with GoTo). As reviewer ZfbG points out, our method does not require “domain knowledge about subgoal formulation/proposal/completion judgment”. We only rely on the weak and domain agnostic assumption of recognizing nouns and adjectives.
>
> Moreover, even if both ELLA and EAGER use demonstrations, EAGER is robust and the QA module can be trained using noisy demonstrations. Indeed, as explained in the supplementary Wide distribution of trajectories (Part C), the bot that generates example trajectories in EAGER can perform a random action at each step with probability p. Furthermore,  EAGER is robust to the success rate of the QA, suggesting that we can reduce the number of training trajectories even more (see in the supplementary Relation between sample efficiency and success rate of the QA (Part C)).
>
> Finally, with an extended time budget for training, it appears that EAGER outperforms ELLA in all the tasks. Changes have been made in the paper to reflect those new observations.
>
> ### 2) Comparison with behavioural cloning and offline RL
>
> The reviewer asks if it is possible to train an agent using BC and the data used to train the QA.
>
> In order to check whether the trajectories generated for training the QA can be used to train an agent using behavioural cloning (BC), we conducted a detailed experiment in the supplementary material Comparison to behavioural cloning (line 96). We conclude that there are not enough example trajectories (moreover the ones given are noisy) to successfully perform BC even for simple tasks like PutNext. Experiments performed in the BabyAI paper [1] show that tasks like PutNextTo-Local require at least 244.000 example trajectories to be successfully learned. We have also trained some agents with offline RL using the IQL method [2] but without success. Once again this is due to the lack of example demonstrations.   We had to design a straightforward GoTo task where this method obtains a non-zero score on the test set (it gets 6%) to make sure all the zero scores were not due to a bug.
>
> ### 3) Comparison to RIDE
>
> The reviewer considers that the fact that EAGER outperformes RIDE is not surprising.
>
> EAGER extracts information from language instructions to perform reward shaping. We compare our method to RIDE, an exploration method that does not use language and addresses sparse reward problems by rewarding impactful change in states. We agree with the reviewer that the better performance of EAGER compared to RIDE is not surprising as the EAGER agent receives some indications based on example trajectories through the QA module. We still maintain this comparison as a baseline for the best method without expert knowledge. Following the indication of the reviewer, we clarified that point in the paper. Yet, due to the low number of examples provided to EAGER, neither BC nor offline RL can be used to warm start an agent before using RIDE: thus, providing the same dataset of prior demonstrations to RIDE would not improve RIDE’s performance.

---

> > ### Author Response · Authors · 2022-08-02
> > **Response to reviewer piZu: clarification on the comparison with ELLA and the use of a dataset (2/2)**
> >
> > ### 4) QG module
> >
> > The reviewer is concerned by the simple structure of the QG.
> >
> > As the reviewer pointed out, our approach to question generation is simple yet general as it could be applied in principle to a wide diversity of contexts, and our aim in this paper is to show a proof of concept for LC-RL. Thus the answer to a question generated by the QG is one of the words from the linguistic instruction. With this approach, EAGER demonstrates that a simple QG can be sufficient to already obtain improvement or results on par with the SOTA (ELLA) but requiring less human intervention. As we mention in the conclusion, more complex QG modules are expected to perform even better, especially in environments where the goals are less descriptive.
> >
> > ### 5) Clarification of input values in BabyAI
> >
> > The reviewer asks for some clarification about the visual input value in BabyAI.
> >
> > About the 3 input values in Line 245, the 3 inputs are 3 integers: one representing the shape of the object, one its color, and one its state. For instance, (4, 1, 1) represents a closed green door. We have updated the manuscript to clarify this.
> >
> > [1] Maxime Chevalier-Boisvert, Dzmitry Bahdanau, Salem Lahlou, Lucas Willems, Chitwan Saharia, Thien Huu Nguyen, and Yoshua Bengio. Babyai: A platform to study the sample efficiency of grounded language learning. In International Conference on Learning Representations (ICLR), 2019.
> >
> > [2]Ilya Kostrikov, Ashvin Nair, Sergey Levine Offline Reinforcement Learning with Implicit Q-Learning, 2021

---

> > > ### Author Response · Authors · 2022-08-08
> > > **Follow-up to Reviewer piZu**
> > >
> > > Thank you again for your review. We understand that this review format is taxing, which is why we have endeavoured to respond clearly to all the points in your review. We hope that if you have any further questions or comments, we can work together to answer them and make this the best paper possible!

---

> > > > ### Comment · Reviewer_piZu · 2022-08-09
> > > > **Updated review**
> > > >
> > > > Thank you for your detailed response.  Since additional experiments are conducted to show the lack of enough example trajectories for BC to work well, I've updated my score to weak accept.   The advantage of EAGER over BC is more clear now.  The robustness, as described in "EAGER is robust and the QA module can be trained using noisy demonstrations", is also a good selling point.

---

> > > > > ### Author Response · Authors · 2022-08-09
> > > > > **Follow-up to Reviewer piZu**
> > > > >
> > > > > We are pleased to hear that our response has addressed your questions and concerns. Thank you again for taking the time to respond.

---

### Author Response · Authors · 2022-08-02
**General response to all reviewers**

We want to thank the reviewers for their thorough evaluation of our work. We are glad that they all expressed interest in our method. Based on their comments and suggestions, we have uploaded a revised version of the manuscript. The individual answer to each reviewer is self-contained, but here we answer the common concerns that several reviewers have expressed.

A common comment in several reviews was about the pre-training process. As we have detailed in our responses and in the paper, the training process of the QA does not undermine the claim that EAGER uses less “expert knowledge” than ELLA.

First of all, ELLA needs human experts that have to determine which class of auxiliary objective has to be associated with the task. The point of EAGER is to remove this human intervention as EAGER is able to extract auxiliary objectives and determine their completion coupled with the associated intrinsic reward through the QG and QA modules. As Reviewer ZfbG points out, our method does not require “domain knowledge about subgoals”.

Our second point is that even if both ELLA and EAGER use demonstrations, EAGER is robust and can deal with noisy data. Indeed, as explained in the paragraph Wide distribution of trajectories in the supplementary (Part B), the bot providing pre-training data in EAGER can take a random action with probability p at each step. Thus trajectories used in EAGER are neither optimal nor efficient, they only successfully complete the task possibly with very suboptimal trajectories. We also showed in the supplementary Comparison with behavioural cloning and offline RL (Part C) that training either behavioural cloning or offline RL based on the  7.500 noisy trajectories used by our QA results in a success rate of 0. Thus, our method is robust and only needs a few non-optimal trajectories to be successful.

To address all the suggestions of the reviewers, the main modifications to our submission are the following:
- We have expanded the problem statement to better explain the role of the QA module and how it is trained, and better stress the relationship to behavioural cloning.
- We have clarified our comparison to behavioural cloning and added a comparison with offline RL.
- We have run the Sequence task with an increased time budget resulting in EAGER getting better results than ELLA, (EAGER outperforming ELLA the previous SOTA on all the tasks).
- Following the advice of Reviewer Pizu, we have added a table listing all the assumptions and expert knowledge required for RIDE, EAGER, and ELLA, see section A in the revised supplementary material.
- As requested by Reviewer ZfBg we have added the PPO baseline without reward shaping and we have added an ablation experiment about policy invariance in the supplementary (Part C).

In the revised version, most of our changes and additions have been put into the supplementary material, due to lack of space. If the paper is accepted, we will get an additional page and make profit of it to move some of the content (at least the table requested by reviewer Pizu) to the main paper. All the revised parts have been colored in red to help the reviewers focus on the modifications.

---

> ### Author Response · Authors · 2022-08-05
> **A summary of the discussions at the attention of the AC (and the reviewers)**
>
> The authors thank again the reviewers for their time and valuable reviews. In addition, we would like to highlight that the discussion period will end Tuesday. We hope that we have covered the reviewers’ concerns. It would be helpful to know their reaction to our response while there is still time to engage in discussion. The authors would really appreciate it if the reviewers could comment on the authors’ responses and raise any remaining concerns.
>
> ### Summary of the main changes to the paper
>
> - We have expanded the problem statement to better explain the role of the QA module and how it is trained, and better stress the relationship to behavioural cloning.
> - We have clarified our comparison to behavioural cloning and added a comparison with offline RL.
> - We have run the Sequence task with an increased time budget resulting in EAGER getting better results than ELLA, (EAGER outperforming ELLA the previous SOTA on all the tasks).
> - Following the advice of Reviewer Pizu, we have added a table listing all the assumptions and expert knowledge required for RIDE, EAGER, and ELLA, see section A in the revised supplementary material.
> - As requested by Reviewer ZfBg we have added the PPO baseline without reward shaping and we have added an ablation experiment about policy invariance in the supplementary (Part C).

---

### Meta-Review · Area_Chair_ncy8 · 2022-08-26

**Recommendation:** Accept
**Confidence:** Certain

**Metareview:**

This paper presents a language-guided auxiliary reward mechanism based on generating Q&A pairs based on agent trajectories and rewarding the agent for producing trajectories that yield correct answers from an answering model. The reviewers broadly found the paper compelling and convincing, and thus I am happy to follow their general consensus in recommending acceptance.

**Award:**

No

---

### Decision · Program_Chairs · 2022-09-14

Accept